# Exploring the neurogenic differentiation of human dental pulp stem cells

**Arwa A. Al-Maswary**[1]*, **Molly O'Reilly**[2], **Andrew P. Holmes**[2], **A. Damien Walmsley**[1], **Paul R. Cooper**[1,3], **Ben A. Scheven**[1]*

1 School of Dentistry, Institute of Clinical Sciences, College of Medical and Dental Sciences, University of Birmingham, Birmingham, United Kingdom, 2 Institute of Cardiovascular Sciences, College of Medical and Dental Sciences, University of Birmingham, Birmingham, United Kingdom, 3 Faculty of Dentistry, Sir John Walsh Research Institute, University of Otago, Dunedin, New Zealand

* AAA680@alumni.bham.ac.uk, arwa2008.dent@gmail.com (AAA-M); b.a.scheven@bham.ac.uk (BAS)

**Data Availability Statement:** All relevant data are within the article and its Supporting Information files.

## Abstract

Human dental pulp stem cells (hDPSCs) have increasingly gained interest as a potential therapy for nerve regeneration in medicine and dentistry, however their neurogenic potential remains a matter of debate. This study aimed to characterize hDPSC neuronal differentiation in comparison with the human SH-SY5Y neuronal stem cell differentiation model. Both hDPSCs and SH-SY5Y could be differentiated to generate typical neuronal-like cells following sequential treatment with all-trans retinoic acid (ATRA) and brain-derived neurotrophic factor (BDNF), as evidenced by significant expression of neuronal proteins βIII-tubulin (TUBB3) and neurofilament medium (NF-M). Both cell types also expressed multiple neural gene markers including growth-associated protein 43 (GAP43), enolase 2/neuron-specific enolase (ENO2/NSE), synapsin I (SYN1), nestin (NES), and peripherin (PRPH), and exhibited measurable voltage-activated Na$^+$ and K$^+$ currents. In hDPSCs, upregulation of acetylcholinesterase (ACHE), choline O-acetyltransferase (CHAT), sodium channel alpha subunit 9 (SCN9A), POU class 4 homeobox 1 (POU4F1/BRN3A) along with a downregulation of motor neuron and pancreas homeobox 1 (MNX1) indicated that differentiation was more guided toward a cholinergic sensory neuronal lineage. Furthermore, the Extracellular signal-regulated kinase 1/2 (ERK1/2) inhibitor U0126 significantly impaired hDPSC neuronal differentiation and was associated with reduction of the ERK1/2 phosphorylation. In conclusion, this study demonstrates that extracellular signal-regulated kinase/Mitogen-activated protein kinase (ERK/MAPK) is necessary for sensory cholinergic neuronal differentiation of hDPSCs. hDPSC-derived cholinergic sensory neuronal-like cells represent a novel model and potential source for neuronal regeneration therapies.

## Introduction

Over recent decades, stem cells have gained special attention for potential nerve regeneration to treat nerve injuries or defects [1,2]. Clinical evidence suggests that current therapies offer limited functional nerve recovery [3] and there are other drawbacks with graft procedures such as nerve sacrifice and nerve mismatch [1]. In dentistry, for example in regenerative

**Funding:** This project was funded by IDB Merit Scholarship (IDB No. 600031755) to A.A.A and University of Birmingham – School of Dentistry (grant No. GAM2271) to B.A.S, A.D.W, P.R.C. The funders had no role in study design, data collection and analysis, decision to publish, or preparation of the manuscript.

**Competing interests:** The authors have declared that no competing interests exist.

endodontics, there is a need for nerve regeneration to achieve functional pulp regeneration [4,5] which regulates pulpal blood flow, defense, and reparative process [6,7].

Stem cells have the potential to differentiate into multiple cell types with neural stem cells giving rise to neuronal cells and their supporting cells "glial and Schwann cells" [8,9]. As a result, the stem cells have been promoted as neuronal cell replacements for nerve repair and regeneration [10,11]. These stem cells can be either transplanted alone [12] or as part of designed engineered tissue/conduit to replace the defective neuronal tissue [13,14]. Stem cell transplantations have demonstrated positive therapeutic nerve regeneration, functional recovery, and neuronal survival in several neurological traumas such as brain injury [15] and spinal cord injury/transection [16,17], optic nerve crush [18], and injured peripheral nerves [12]. However, the use of stem cells in transplantation therapies may be limited by the small populations differentiated into neuronal cells or undesired cell proliferation or differentiation [19,20]. Consequently, it has been proposed that the use of ready *in vitro* differentiated cells derived from stem cells is more promising for *in vivo* nerve regeneration [21,22]. For example, some studies demonstrated that transplantation of the pre-differentiated stem cells into a neuronal phenotype "neuronal cell models" results in greater restoration of neuronal loss [21], enhances nerve regeneration and functional recovery in brain [21,23], spinal cord [24], and peripheral nerve injuries [25,26]. Interestingly, it has also been reported that neuronally differentiated stem cells secrete greater amounts of neurotrophic factors [25,27]. Hence, these neuronal cell models are not only neuronal cell replacements for the neuronal injury or defect, but they may further boost the nerve regeneration via their neurotrophic secretions. Neuronal cell models derived from stem cells are also useful for *in vitro* studies in neuroscience [28,29]. For example, these neuronal cell models can be used to study neurodegenerative diseases such as Parkinson's disease [30] and Alzheimer [31], pharmacological-related topics "drug discovery and toxicity testing" [32,33], and neurodevelopment and injury [28]. As these neuronal cell models are differentiated from primary stem cells, they are more appropriate for simulation of the physiological properties of *in vivo* neurons [34,35].

Dental pulp stem cells (DPSCs) are primary ecto-mesenchymal stem cells (MSCs) that have gained attention as a potential source for neuronal regenerative therapies. The neurogenic potential of DPSCs is closely related to their embryonic origin and various biological characteristics. DPSCs are derived from cranial neural crest cells during tooth development [36,37]. In this context, it has been demonstrated that DPSCs retain the properties of neural crest cells such as EphB/Ephrin-B molecules and Wnt1-marker in *in vitro* cell culture which possess the differentiation capacity into any neural crest-derived tissue, including neuron [38,39]. In addition to the stem cell markers, the expression of the neural markers in non-differentiated DPSCs, such as musashi12, nestin, MAP2ab, βIII-tubulin, N-tubulin, and neurogenin-2 underline their potential for neuronal differentiation [40,41]. Furthermore, DPSCs express neurotrophic factors such as NGF, GDNF, BDNF, and NT-3 which are demonstrated to have neurogenic and cell survival effects [42,43]. Moreover, DPSCs have been considered to be able to differentiate into specific neuronal cells of nervous system depending upon the induced environment [44,45] which make DPSCs an attractive cell source for specific neuronal-lineage regeneration therapies. Additionally, DPSCs exhibit other favorable non-neurogenic factors such as their unique immunomodulation properties which prevent the possibility of immune rejection/reactions [46,47] or tumor formation [48,49] which is reported in other stem cell transplantations [50,51]. Finally, DPSCs are easily obtainable from teeth extracted for various dental reasons without raising ethical concerns [52]. The neuro-regenerative potential of hDPSCs have been highlighted for dental pulp regeneration [53,54], retinal [55,56] and nerve injury [57,58]. Thus, DPSCs potentially offer a safe neurogenic-potential stem population suitable for multiple clinical neuronal therapeutic applications.

Different methods have been described to differentiate human DPSCs (hDPSCs) into neuronal-like cells. Most differentiating protocols for hDPSCs use complex mixture of supplements either in multiple stages [40,45,59,60] or/and long culture duration, "more than a month" [61–66], which make the procedures relative expensive and time-consuming. On the other hand, one protocol reported that serum-free media without any supplementations can differentiate mice DPSCs into neuronal-like cells and expressed MAP2, nestin, and Tub3/βIII-tubulin neuronal markers [67]. However, these neuronal markers have been reported in non-differentiated DPSCs [41,68] which may not provide sufficient evidence for neuronal differentiation, particularly with no functional testing. Furthermore, this serum-free protocol has been recently used by Madanagopal et al., [69] as one of three protocols to differentiate hDPSCs into neuronal cell type. This study reported that the serum-free media alone did not result in neuronal differentiation of hDPSCs compared with what has been reported in mice DPSCs by Zainal et al., [67] and the authors interpreted that this may occur due to the genetic and physiological differences between mice and human [69,70]. In addition, the concept of neuronal differentiation in serum-free media without supplementations is previously discussed by Croft and Przyborski [71] who reported that culturing in serum-free media is an environmental culture stressor which results in pseudo-neuronal morphology and expression "artifacts". Moreover, there is little convincing evidence for successful functional DPSC neurogenic differentiation [61,62]. Hence, there is a need for simple, and relatively rapid differentiating protocol underpinned with sufficient evidence for neuronal differentiation and functionality.

A straightforward 2-component method or sequential supplementation of all-trans retinoic acid: (ATRA) and then brain-derived neurotrophic factor (BDNF) was described by Encinas et al. [72] to differentiate human SH-SY5Y neuroblastoma cells into a mature neuronal cells. This method has been well-investigated and documented to produce mature, functional, cholinergic neuronal cell types of the human SH-SY5Y neuroblastoma cells [73–75]. In this study, we adopted this method to differentiate hDPSCs for the first time in parallel with SH-SY5Y cells as control cells because they are successfully differentiated by the sequential supplementation. In addition, SH-SY5Y cells are widely used as human neural stem cells to produce models for neuroscience studies such as related Alzheimer and Parkinson's diseases [76,77], energetic neuronal vulnerability [78], neurotoxicity [79,80], and testing of MSC paracrine effects for neuronal differentiation and matuation [81,82].

Although, there are many signaling pathways active in the nervous system, mitogen-activated protein kinase (MAPK) and phosphatidylinositol-3-Kinase/protein kinase B (PI3K/Akt) are reportedly central and essential in the overall regulation of neural differentiation and survival [83,84]. Previous work has also demonstrated that the Extracellular signal-regulated kinase/Mitogen-activated protein kinase (ERK/MAPK) pathway induces neuronal differentiation whereas the PI3K/Akt pathway maintains cell survival of SH-SY5Y and bone marrow MSCs during the neuronal differentiation [75,85]. Furthermore, the involvement of ERK/MAPK pathway is reported in axonal outgrowth and peripheral nerve regeneration [86,87].

The aim of this study was to explore and characterise a novel neuronal model using hDPSCs based on the established SH-SY5Y neurogenesis model and to investigate whether the ERK/MAPK pathway is involved in the hDPSC neuronal differentiation process.

## Materials and methods

### Cell culture

Human DPSCs were obtained from two suppliers (#PT-5025, Lonza, Slough, UK and #ax3901, Axol, Cambridge, UK; suppliers' information regarding the stem cell characterization is provided in **S1 Table**). Cells were cultured in alpha-modified minimum essential medium (α-

MEM) (Biosera, UK) supplemented with 2 mM L-glutamine, 10% fetal bovine serum (FBS) (Biosera, UK) and 1% penicillin/streptomycin (100 IU.ml−1). The SH-SY5Y neuroblastoma cells (ATCC® CRL-2266™, USA) were cultured in Dulbecco's modified Eagle's medium/ Ham's nutrient mixture F12 (DMEM/F12) (Sigma Aldrich, UK) supplemented with 2 mM L-glutamine, 10% FBS and 1% penicillin/streptomycin (100 IU.ml−1) in parallel with hDPSC culture used as a positive control. Cultures were incubated in a humidified atmosphere at 37˚C and 5% $CO_2$ until reaching ~ 80% confluency before neurogenic differentiation induction. The medium was exchanged every 2–3 days. All experiments were conducted at passages of 2–4 using the hDPSCs and of 17–21 using the SH-SY5Y cells. Further details about cell culturing of both cell types are described in the **S1 File**.

## Neurogenic differentiation of hDPSCs

Neurogenic induction was conducted as described by Encinas et al. [72] with minor modifications (i.e., 10% FBS instead of 15% FBS and DMEM/F12 media instead of DMEM). Firstly, cells were seeded on collagen-I coated surfaces of 6-well plate (Thermo Fisher Scientific, UK) or laminin/collagen-coated coverslips (Electron Microscopy Sciences, UK) for the microscopy studies. Whereas the cells were used for real-time PCR seeded on uncoated T25 flasks (Thermo Fisher Scientific, UK) for preserving the purity of RNA and getting higher amount of RNA. The seeding density was 5000 cells/cm$^2$ (SH-SY5Y), and 625 cells/cm$^2$ (hDPSCs) based on preliminary studies. The seeded cells were incubated overnight to allow cells to attach to the culturing surface before conducting the differentiation experiment. Subsequently, the differentiation and control media were freshly prepared for each experimental group as described in **Table 1** (all supplements were defrosted and immediately used for the experiment to avoid the material degradation over the time). After that, the overnight media were replaced with ATRA-supplemented (R 2625, Sigma-Aldrich, UK) or with control media and then incubated in a humidified incubator at 37˚C and 5% $CO_2$. This media change step was performed in limited light in the laboratory room and the culture hood's light was switched off due to the light-sensitive nature of ATRA. Then, the media were changed after 2–3 days with fresh media with or without ATRA as previously highlighted in **Table 1** and then incubated in a humidified incubator for additional 2–3 days. After 5 days of treatment with ATRA, all experimental groups were washed twice with blank media without any supplementations to remove the FBS and ATRA remnants in the cell culture before the second "BDNF" stage for the ATRA→BDNF and ATRA→0% serum groups. Subsequently, the designed experimental ATRA→BDNF group received BDNF supplementation (78005, STEMCELL TECHNOLOGIES; SRP3014,

**Table 1. Experimental groups (differentiated and control groups).**

| Experimental groups | Culturing medium and supplementations | Incubation time (days) | Total (days) |
|---|---|---|---|
| Control (standard cell culture) | 10% FBS DMEM/F12* supplemented with 10 μM DMSO$. | 12 | 12 |
| ATRA | 10% FBS DMEM/F12* supplemented with 10 μM ATRA. | 12 | 12 |
| ATRA→BDNF | 1st stage: 10% FBS DMEM/F12* supplemented with 10 μM ATRA. | 5 | 12 |
| | 2nd stage: Serum-free DMEM/F12* supplemented with 50 ng/ml BDNF. | 7 | |
| ATRA→ 0 serum (2nd control) | 1st stage: 10% FBS DMEM/F12* supplemented with 10 μM ATRA. | 5 | 12 |
| | 2nd stage: Serum-free DMEM/F12* without any supplement. | 7 | |

* Supplemented with penicillin/streptomycin.

$ DMSO is added as it is the dissolvent used to prepare the ATRA, so the control group is identical to differentiating group but without the differentiating supplement "ATRA".

Sigma-Aldrich, UK) in serum-free media whereas its control group (ATRA→0% serum) received only serum-free media. The ATRA→0% serum group (identical group to ATRA→BDNF but with absence of BDNF) was added to control the presence of BDNF in the ATRA→BDNF group and determine if the absence of BDNF would result in the same outcomes. The other two experimental groups (control and ATRA) were continued culturing in 10% FBS DMEM/F12 media with and without ATRA as previously described. All cell culture groups were incubated in a humidified incubator till the next media change. Finally, the subsequent media change was performed after 3–4 days prior to the end of the neurogenic induction period (12 days). For more details regarding preparation, diluting the differentiating supplements and culturing, see the **S1 File**.

## Immunocytochemistry

The cells were fixed for 10 min with 4% paraformaldehyde in PBS (Alfa Aesar, UK) and then gently washed twice to remove the remaining fixative solution. The blocking step was then performed for 1h with blocking solution (10% goat serum, 3% bovine serum albumin BSA, and 0.1% Triton X-100 (Sigma, UK) were prepared in PBS). Subsequently, the blocking agent was removed, and the diluted primary antibody was applied and incubated overnight at -4°C. Cells were then washed with PBS (3x10min) and incubated with the secondary antibodies for 1h. Afterward, the cells were gently washed (3x10min) and mounted on microscopy slides using aqueous mounting media containing DAPI (Abcam, UK). The primary and secondary antibodies stains were prepared in a diluent buffer (3% BSA and 0.05% tween-20 in PBS). All information regarding the antibodies and dilutions are provided in **S2 Table**. The primary antibody application step was omitted in the negative control groups and cells were incubated with diluent buffer alone instead. Images were captured under 40x-oil lens magnification using confocal microscopy (Zeiss LSM 700 confocal microscope, Germany).

## Quantitative real-time polymerase chain reaction (RT-qPCR)

RNA was extracted using the Qiagen RNeasy Mini kit according to the manufacturer's instructions. RNA purity and concentration were determined using a Spectrophotometer (BioPhotometer Plus, Eppendorf, Germany) at an absorbance of 260/280nm. RNA integrity was visualized using agarose gel electrophoresis. Subsequently, cDNA was synthesized from 1 μg RNA using the Tetro cDNA Synthesis Kit (Bioline, UK). The cDNA was amplified by qPCR using the LightCycler® 480 SYBR Green I Master kit (Roche, UK). The qPCR cycling protocol was run as described by Forster et al. [78] with minor modifications using the Roche LightCycler® 480 II machine PCR system. All samples were run in duplicate or triplicate wells with two negative controls "no cDNA-RNase free water" per each primer pair in every PCR which were run to control for genomic DNA contamination. The melting curve was also checked for each product and selected PCR products were further analyzed by agarose gel electrophoresis to confirm size. The crossing point data (Cp) of the gene expression were computed by the LightCycler 480 software using the fit-points methods according to the manufacturer's instructions.

The efficiencies of all primers were validated for Real-time PCR as previously described and recommended by Pfaffl [88]. The efficiency values were logarithmically calculated using LightCycler® 480 software by creating standard efficiency curves for the serial dilutions of each primer. The stability of the four housekeeping references (GAPDH, RPLA13, HPRT1, and B2M) was also investigated to select the most stable references for normalization of qPCR data as described by Pfaffl et al. [89] and Li et al. [90]. The selection of the most stable housekeeping reference gene was computed by a statistical algorithm analysis program "Normfinder" [91]

which determined the HPRT1 as the most stable reference gene amongst the housekeeping references. The fold-change gene expression was calculated as described by the Pfaffl method [88] which includes the Cp data of the most stable reference gene (HPRT1) and efficiency values of both housekeeping and gene of interest primers. The primers, their related information, and efficiency values are provided in **S3 Table.**

## Electrophysiological recordings of whole-cell sodium and potassium currents

For $Na^+$ current recordings ($I_{Na}$), cells were superfused at 3ml/min-1, 22±0.5˚C with a solution containing in mM: NaCl 145, KCl 4.5, HEPES 10, $NiCl_2$ 2, $CaCl_2$ 1.8, $MgCl_2$ 1.2 and glucose 10, pH 7.4 (CsOH) as described [92,93]. The internal pipette solution in mM was: CsCl 115, NaCl 5, HEPES 10, EGTA 10, MgATP 5, $MgCl_2$ 0.5 and TEA 20, pH 7.2 (CsOH). Whole-cell patch-clamp recordings were obtained in voltage-clamp mode using an Axopatch 200B amplifier. Tip resistance was 1.5-3MΩ and cells were stimulated at 1 Hz. Current-voltage relationships were examined using 100 ms step depolarizations to test potentials ranging from -40 mV to +60 mV from a holding potential of -100 mV.

For recordings of $K^+$ currents ($I_{Kss}$), the same external solution was used but the internal solution contained in mM: KCl 135, NaCl 5, EGTA 10, HEPES 10, MgATP 3, $Na_3GTP$ 0.5 and glucose 5 (pH 7.2, KOH), as described [94,95]. $K^+$ currents were evoked by step depolarizations (500 ms) to test potentials between -60 mV and +40 mV from a holding potential of -70 mV, at 1 Hz pacing. Both $I_{Na}$ and $I_{Kss}$ were normalized to cell capacitance.

## Role of the ERK1/2 signaling in the neurogenic induction

The cells were initially differentiated with ATRA-supplemented media for 5 days. Cells were then washed twice with blank media and then cultured in serum-starved media for 5h. Subsequently, the cells were pre-treated with or without 10 μM of ERK/MEK inhibitor (U0126, Cell Signaling Technology, USA) for 1h prior to BDNF supplementation. The BDNF incubation was performed at either 5 min to quantify the ERK1/2 phosphorylation by ELISA or at 48h to assess the effect on differentiation by immunocytochemical expression of mature neuronal marker (Neurofilament medium: NF-M) in the presence or absence of the inhibitor.

## Phospho-ERK/MAPK quantification by Enzyme-linked immunosorbent assay (ELISA)

Cell lysis and ELISA procedures were performed following the manufacturer's instructions (PathScan® Phospho-p44/42MAPK (Thr202/Tyr204), Cell signaling Technology, USA). The cells were lysed with lysis buffer (Cell Signaling Technology, USA), and the supernatants of the lysed cells were used for ELISA. The cell lysates of the samples were incubated in the phospho-p44/42MAPK (ERK1/2)-coated 96-well plate overnight at 4˚C. Afterward, the sequential incubation with the detection antibody, the HRP-linked secondary antibody, and TMB substrate was for 1h, 30 min, and 10 min at 37˚C, respectively. Generous washing (4 times) with wash buffer was performed after each step using an automated plate washer (Bio-Tek Instruments, USA). Finally, the assay reaction was terminated by adding stop solution and reading the absorbance at 450 nm using Spark microplate reader (Tecan Trading AG, Switzerland).

## Statistical analysis

Statistical analysis of the data was performed using SPSS Statistics version 26 and 27 (IBM, USA). The significance level was set at P< 0.05. The groups were compared by Kruskal-Wallis

test with a pairwise comparison unless otherwise stated. Significance values were adjusted by Bonferroni correction for multiple tests. The plotting data in the graphs were presented as mean ± SD unless otherwise stated. All experiments were repeated at least twice. Graphs were generated by the GraphPad Prism 9 software package (GraphPad, San Diego, CA, USA).

## Results

### hDPSCs acquired neuronal-like morphological features after neuronal induction

The hDPSCs and SH-SY5Y cells demonstrated comparable results in which the cells exhibited the highest morphological transition into neuronal-like cells after the sequential supplementation method (ATRA→BDNF) (Fig 1A–1L). SH-SY5Y cells exhibited fine neurite extensions after ATRA supplementation stage (Fig 1B). These neurite extensions became more extensive and branched after the BDNF supplementation stage which resulted in multipolar neuronal-like morphology with network communications (Fig 1D). The parallel control group in the absence of BDNF and FBS supplementation (ATRA→0% serum) exhibited loss of the cells, neurite phenotype and extensions previously produced by the ATRA supplementation (Fig 1C). Whereas hDPSCs began to change morphologically into neuronal-like features after ATRA supplementation stage by exhibiting a bipolar elongation appearance (Fig 1F, solid arrows) compared with the control group (Fig 1E). Subsequently, the hDPSCs acquired more

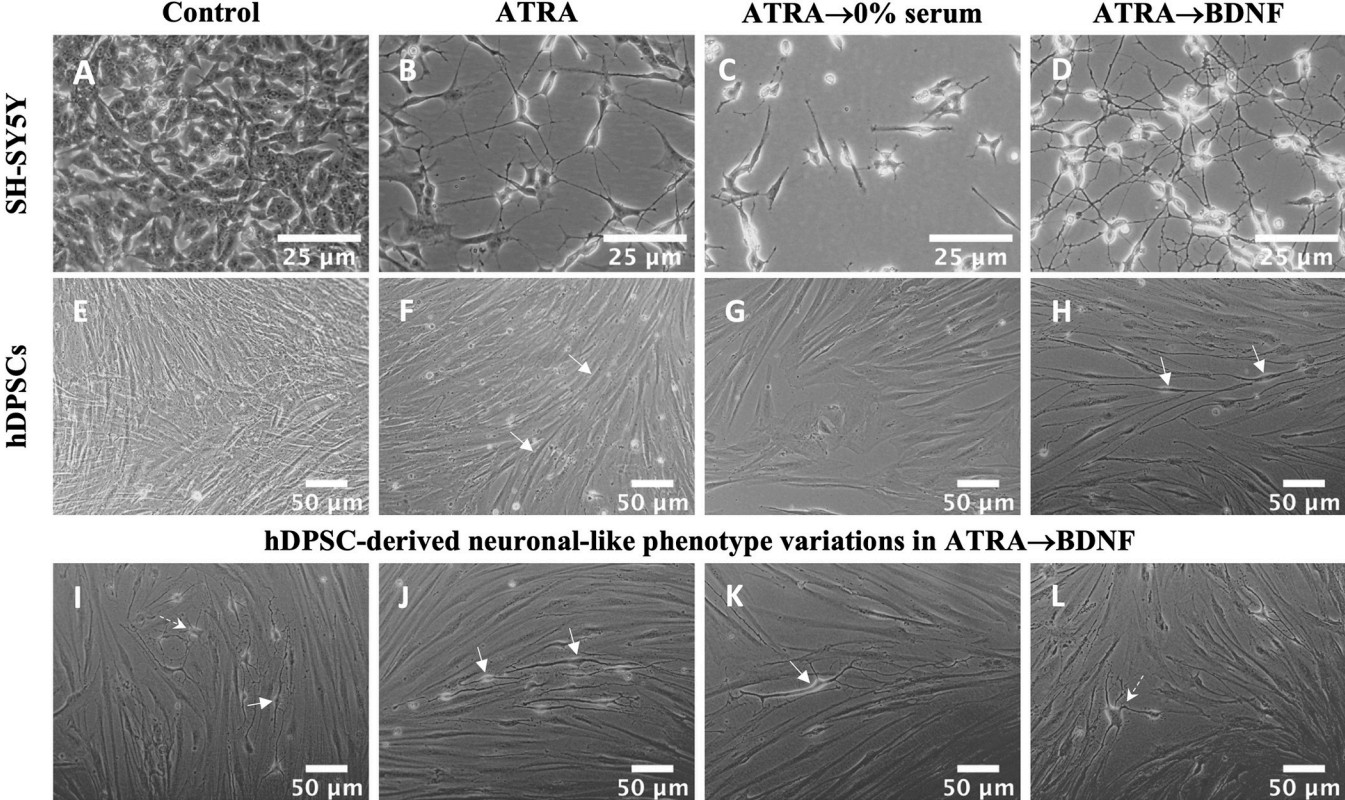

**Fig 1. Neural differentiation of SH-SY5Y and hDPSC experimental groups.** (A-D) Phase contrast images of SH-SY5Y groups; (A) control, (B) ATRA, (C) ATRA→0% serum, and (D) ATRA→BDNF. (E-L) Phase contrast images of hDPSC groups; (E) control, (F) ATRA, (G) ATRA→0% serum and (H-L) ATRA→BDNF. The phase contrast images were background corrected and converted to 8-bit images. Scale bars are shown. Solid arrows indicate bipolar elongation/morphology; dotted arrows indicate multipolar neuronal-like morphology.

marked and defined neuronal-like features reflected by defined cell bodies displaying bipolar "mainly" and multipolar neuronal-like morphology following the BDNF supplementation stage (Fig 1H–1L; solid arrows indicate bipolar, whereas dotted arrows indicate multipolar neuronal-like morphology). The typical bipolar neuronal-like morphology is presented in Fig 1K and the typical multipolar neuronal-like morphology is shown in Fig 1L. In contrast, the cells of the parallel control group "ATRA→0% serum" did not exhibit the neuronal-like changes when incubated in serum-free media without BDNF supplementation (Fig 1G). These results indicate that the greatest number of neuronal features were induced by the sequential supplementation method "ATRA→BDNF". Subsequently, the experimental groups were tested for neuronal marker expression to reveal their neuro-immunopositivity and lineage identity.

## Induced hDPSCs immunocytochemically expressed neuronal markers

SH-SY5Y cells and hDPSCs differentiated with ATRA alone or with BDNF supplementation expressed increased neuronal cytoskeleton marker (βIII-tubulin: TUBB3) [96] and the mature marker (Neurofilament medium: NF-M) [97,98] with the highest expression being in the ATRA→BDNF group (Fig 2A and 2B). Notably, the NF-M staining was only evident in the bipolar neuronal-like differentiated hDPSCs, whereas the multipolar neural-like cells did not present any NF-M expression (Fig 2B). Cultures were also stained for Glial fibrillary acidic protein (GFAP) to assay for the presence of astrocyte glial-like cells [99]. GFAP was weakly expressed in the control group of SH-SY5Y cells and was reduced in the differentiated groups (Fig 3A). Whereas none of the hDPSC groups expressed GFAP and this included the multipolar glial-like cells (Fig 3B). Subsequently, this neuronal marker profile was further supported by qPCR data which investigated a broader panel of neuronal gene markers.

## Neuronal gene markers were specifically upregulated in hDPSCs reflecting differentiation toward a sensory cholinergic neuronal lineage

A panel of specific neuronal gene markers were assessed by real-time qPCR to confirm the neuronal differentiation and characterize the neuronal lineage. The SH-SY5Y and hDPSC data showed that the sequential supplementation protocol (ATRA→BDNF) resulted in a greater number of neuronal gene markers expressed compared with ATRA alone supplementation (Fig 4). In SH-SY5Y cultures, both supplementation methods (ATRA alone and ATRA→BDNF) induced significant gene upregulation of several neuronal markers, including Enolase 2/neuron-specific enolase (ENO2/NSE), Nestin (NES), Peripherin (PRPH), Acetyl-cholinesterase (ACHE), Choline O-acetyltransferase (CHAT), and Sodium channel alpha subunit 9 SCN9A/Na$_v$1.7 (Fig 4A–4D). The ATRA→BDNF method stimulated significant gene expressions of additional neuronal markers: Growth-associated protein 43 (GAP43), and Synapsin I (SYN1) (Fig 4A and 4B). In hDPSCs, ATRA alone and ATRA→BDNF methods triggered significant gene upregulations of ENO2/NSE, SYN1, CHAT, and SCN9A/Na$_v$1.7 (Fig 4E–4H). Notably, GAP43, NES, PRPH, ACHE, POU class 4 homeobox 1 (POU4F1/BRN3A) markers were only significantly increased by the ATRA→BDNF supplementation protocol (Fig 4E–4H).

In contrast, both cell types demonstrated a significant reduction in the gene expression of the GFAP (Fig 4A and 4E). The gene levels of Dopamine beta-hydroxylase (DBH) and Motor neuron and pancreas homeobox 1 (MNX1) markers showed no change in the differentiated groups of the SH-SY5Y cells (Fig 4C and 4D). Whereas DBH was not detected in all experimental hDPSC groups and MNX1 was significantly reduced in hDPSC differentiated ATRA→BDNF group (Fig 4G and 4H). These significantly reduced levels or no change in

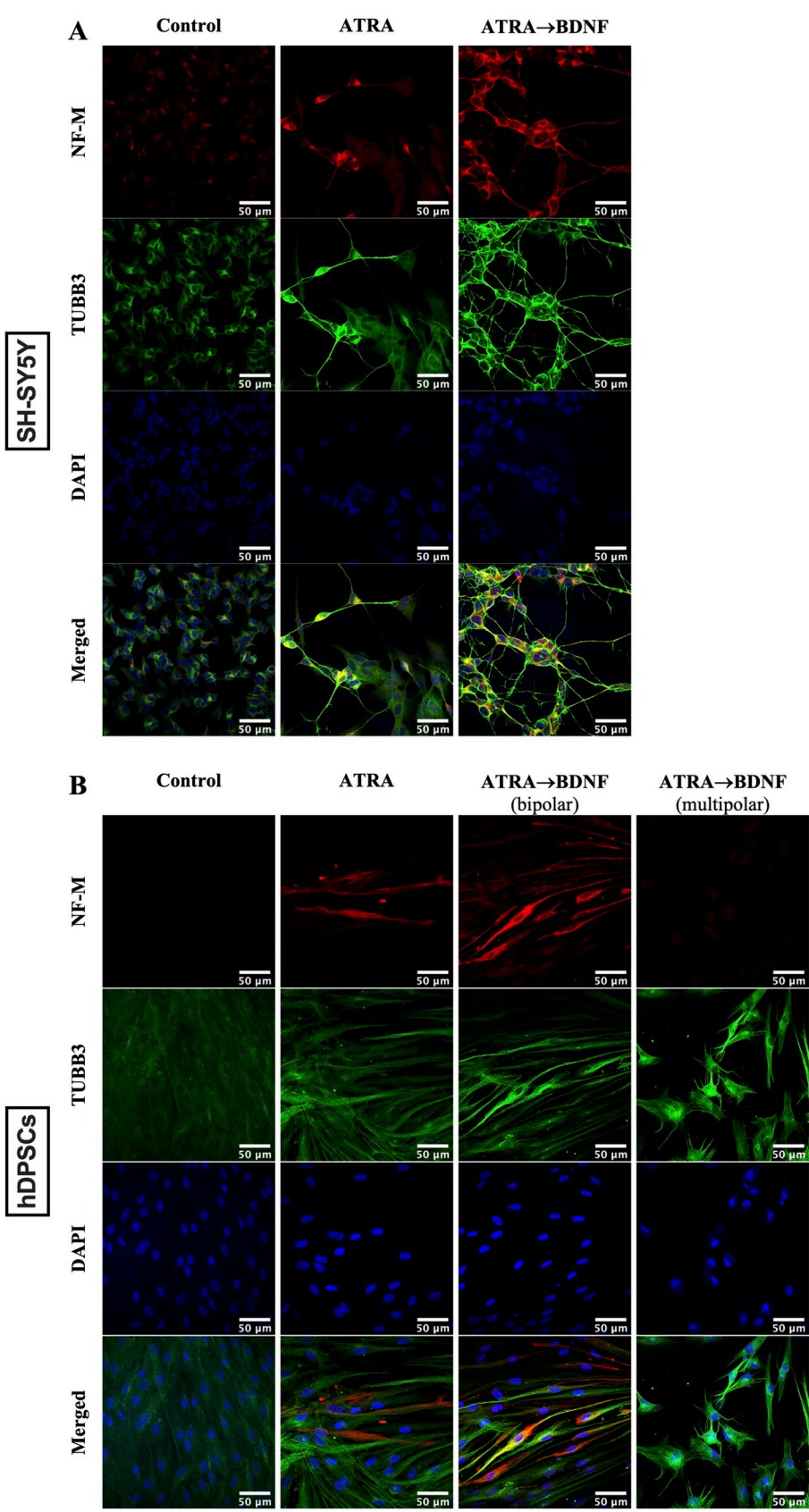

**Fig 2. Immunocytochemical analysis of neuronal markers (NF-M and TUBB3).** (A) SH-SY5Y and (B) hDPSCs. Scale bar is 50 μm in all images.

gene expression of certain specific neuronal markers may confirm the specificity of the resultant neuronal-like cells. For example, significant reduction in astrocyte glial marker (GFAP) [99] with the concomitant upregulation of neuron-specific marker (ENO2/NSE) [100] indicated that the differentiation was induced toward neuronal cells rather than astrocyte glial cells (**Fig 4A and 4E**). No change or significant reduced levels in gene expression of motor marker (MNX1/HLXB9/HB9) [101] and simultaneously increased in gene expression of sensory voltage-gated sodium channel marker (SCN9A/Na$_v$1.7) [102] with or without upregulation of another sensory marker: POU4F1/BRN3A [103] indicate the specialized sensory identity of these neuronal-like cells (**Fig 4D and 4H**). Finally, no change or absence of detection of the noradrenergic neurotransmitter marker (DBH) [104] with the concomitant upregulation of cholinergic neurotransmitter markers (CHAT and ACHE) [105,106] suggests the specialized cholinergic identity of the resultant neuronal-like cells (**Fig 4C and 4G**). The gene expression data support the superiority of the sequential supplementation method and suggest a guided differentiation toward sensory cholinergic neuronal lineage in both cell types.

## Induced hDPSCs demonstrated a significant neuronal electrophysiological profile

To test for functional changes induced by the different treatment protocols, $I_{Na}$ and $I_{Kss}$ were measured in SH-SY5Y cells and hDPSCs. Treatment of SH-SY5Y cells with ATRA and BDNF but not ATRA alone led to a significant upregulation in $I_{Na}$ and $I_{Kss}$ compared with control (**Fig 5A and 5B**). The ATRA→BDNF treated SH-SY5Y cells also had a significantly larger cell capacitance (**Fig 5C**). In hDPSCs, control untreated cells did not display any measurable amount of either $I_{Na}$ or $I_{Kss}$ (**Fig 5D and 5E**). Although some cells treated with ATRA alone had measurable $I_{Na}$, this was highly variable and overall, not significantly different to control cells (**Fig 5D**). However, for ATRA→BDNF treated hDPSCs, peak $I_{Na}$ was larger, more consistently measurable, and significantly elevated compared with control over a range of test potentials between -20 to +40mV (**Fig 5D**). Similarly, $I_{Kss}$ was apparent in some cells treated with ATRA alone and was not significantly different from control cells (**Fig 5E**). However, hDPSCs treated with ATRA→BDNF displayed significant elevations compared with control over a range of test potentials between -10 and +40mV (**Fig 5E**). Measurements of cell capacitance also suggested a significant elevation in cell capacitance in the ATRA→BDNF hDPSCs compared with control (**Fig 5F**).

## ERK1/2 inhibitor blocked neuronal differentiation and ERK1/2 phosphorylation

To determine whether neuronal differentiation using the ATRA→BDNF method depended on the ERK/MAPK signaling pathway, the neuronal differentiation inhibition was assessed by immunocytochemical expression of the mature neuronal marker (NF-M) with and without ERK/MEK inhibitor (U0126). In addition, ELISA was used to quantify the phospho-ERK1/2 levels of the experimental groups in the presence or absence of the ERK/MEK inhibitor (U0126).

The SH-SY5Y cells and hDPSCs presented comparable immunocytochemical expression with BDNF supplementation inducing the greatest levels of NF-M immunostaining among the experimental groups (**Fig 6A and 6B**). In contrast, the BDNF-induced NF-M immunostaining

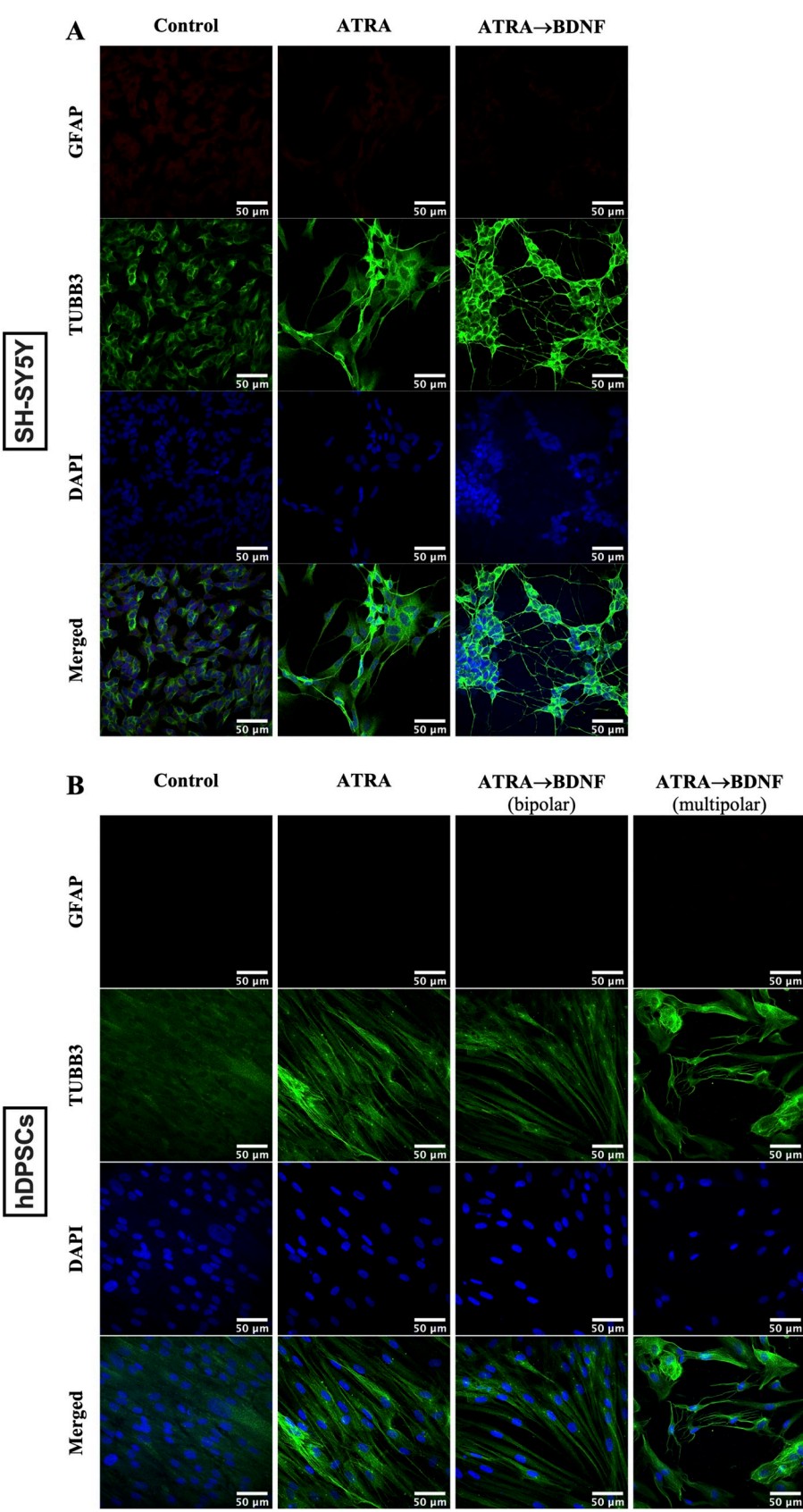

**Fig 3. Immunocytochemical analysis of GFAP in comparison to TUBB3.** (**A**) SH-SY5Y and (**B**) hDPSCs. Scale bar is 50 μm in all images.

increases were markedly reduced in the presence of the U0126 inhibitor (BDNF plus U0126) which demonstrated similar immunocytochemical expression to those of the control group with and without the inhibitor (**Fig 6A and 6B**). These findings suggest the ERK/MEK inhibitor (U0126 inhibitor) completely ablated the differentiating effect of BDNF supplementation.

ELISA data demonstrated that there was a highly significant difference between the experimental groups (One-way ANOVA: $P$ = 0.000, F ratio (df) = 159.196 (3,16), 119.045 (3,16) for SH-SY5Y and hDPSCs, respectively). The BDNF-supplemented groups in both cell types (SH-SY5Y cells and hDPSCs) showed noticeable upregulations of phospho-ERK1/2 levels compared with the control group, albeit the statistically significant increase was only detected in SH-SY5Y cell type (**Fig 7**, SH-SY5Y: $P$ = 0.000 and hDPSCs: $P$ = 0.209). The pre-treatment of the cells with the ERK/MEK inhibitor (U0126) significantly reduced the phospho-ERK1/2 levels induced by BDNF supplementation (BDNF+U0126) compared with BDNF group alone (**Fig 7**, SH-SY5Y: $P$ = 0.000 and hDPSCs: $P$ = 0.001). There was also a significant reduction in control groups with the inhibitor (control+U0126) compared with control groups (**Fig 7**, SH-SY5Y and hDPSCs: $P$ = 0.000). Consequently, the increase of the phospho-ERK1/2 levels in the BDNF-supplemented group compared with those of control group and concomitant reduction of the phospho-ERK1/2 levels in the pre-treated groups with ERK/MEK inhibitor (control+U0126 and BDNF+U0126) suggest the involvement of ERK/MAPK pathway in control and supplemented groups, however BDNF supplementation induced further activation of this pathway. Overall, the concurrent increase of the differentiating effect assessed by immunocytochemical expression of the mature neuronal marker NF-M and phospho-ERK1/2 levels assessed by ELSIA in the BDNF-supplemented groups and their reduction in the presence of the ERK/MEK inhibitor indicate that the ERK/MAPK signaling is involved in regulating neuronal differentiation.

## Discussion

In the present study, hDPSCs were successfully differentiated into neuronal-like cells using the SH-SY5Y sequential neurogenesis supplementation method. This novel and simple approach for establishing a neuronal DPSC differentiation model is supported by microscopic, molecular, and functional evidence. Moreover, this study indicates the sensory cholinergic nature of the differentiated hDPSCs and involvement of ERK/MAPK signaling in the differentiation process.

The ATRA is commonly used to induce neurogenic differentiation of multiple cell lines and stem cells such as SH-SY5Y human neuroblastoma cells [107,108], P19 mouse embryonal carcinoma cell line [109,110], embryonic stem cells [111,112], and mesenchymal stem cells [113]. However, Takahashi et al. [114] and Goldie et al., [74] reported that ATRA alone results in immature neural differentiation of SH-SY5Y and neural stem cells and should be supplemented in combination with neurotrophin such as BDNF to establish full neural maturation. In this context, Encinas et al., [72] and Takahashi et al. [114] reported that sequential induction of ATRA and followed by neurotrophin treatment is critical as the ATRA pretreatment increases the cellular response to neurotrophin (s), including BDNF in SH-SY5Y cells and neural stem cells, respectively. Another study by Bi et al., [115] emphasized that the ATRA pre-induction "activating retinoid signaling" improved neural differentiation of mesenchymal stem cells. Hence, it was hypothesized that the successful neuronal differentiation by sequential

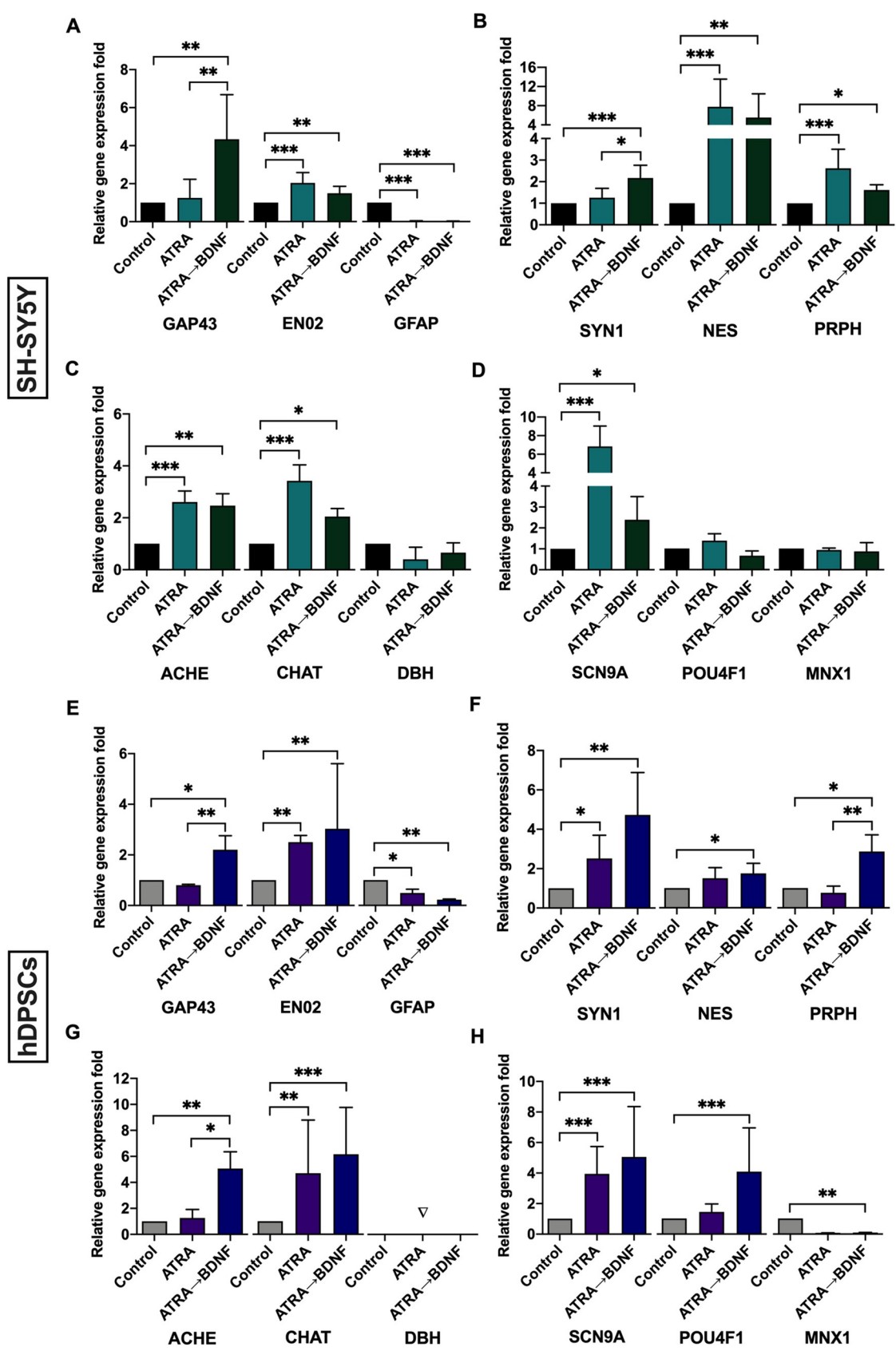

**Fig 4. Quantitative gene expression of specific neuronal markers as determined by real-time qPCR.** (A-D), Experimental groups of SH-SY5Y. (E-H), Experimental groups of hDPSCs. Fold change was calculated using the Pfaffl method and normalized against the most stable housekeeping gene reference HPRT1 (SH-SY5: n = 9, except for ACHE, n = 6, CHAT, POU4F1, and MNX1, n = 3; DPSCs: n = 6, except for EN02, CHAT, SCN9A, and POU4F1, n = 9; Data plotted as mean ± SD; *P< 0.05, **P< 0.01, ***P< 0.001). Note: ▽ indicates not detected.

supplementation of ATRA and then BDNF in SH-SY5Y cells would also result in mature neuronal differentiation of hDPSCs.

Indeed, the differentiated of hDPSCs by the sequential supplementations of ATRA and then BDNF treatment resulted in apparent neuronal morphological features such as phase-bright cell bodies and bipolar or multipolar neurite-like extensions as shown in **Fig 1**. This observation was consistent with the neuronal induction found in the SH-SY5Y cell model and is comparable with previous SH-SY5Y studies [72,74], underscoring the neurogenesis process induced in the hDPSC cultures. The mixed outcome of bipolar and multipolar neuronal-like morphology cells in the differentiated hDPSCs is also reported by Kiraly et al., [40] which indicates the presence of different neuronal phenotypes. Consequently, the mature neuronal marker (NF-M) [97,98] and astrocyte glial marker (GFAP) [99] were used to differentiate between both resultant neuronal-like cells. The bipolar differentiated cells only expressed NF-M and lacked GFAP expression in both neuronal phenotypes "bipolar and multipolar neuronal-like cells". The multipolar differentiated hDPSCs acquired glial-like cell morphology which may represent glial cells, but not the astrocyte glial subtype due to lack of GFAP expression. These findings suggest a mixture of mature bipolar neuronal-like and multipolar glial-like cells but not the astrocyte glial-like cells. This interpretation is in agreement with that of Luzuriaga et al. [116] who reported that BDNF can reprogram hDPSCs into both neurogenic and gliogenic lineages. Hence, this explains the mixed outcome of bipolar and multipolar neuronal-like cells observed here. Indeed, the presence of bipolar and multipolar neuronal-like cells in the same culture supports the concept of heterogeneity of specific markers in hDPSC cultures which results in guiding of the cells toward specific lineages [117,118]. In other words, hDPSC cultures have differences in cellular markers that govern the differentiation ability toward a specific cell type. For example, the DPSCs expressing high nestin were reported to differentiate into neuronal and glial lineages compared with no differentiation of DPSCs with low nestin expression [117]. In this regard, we assume that there are additional specific makers/factors which may induce the individual cell populations in the same hDPSC cultures to differentiate into bipolar or multipolar morphology. Thus, further investigation is required to identify the reason(s) underlying the different outcomes of differentiated neuronal-like cells in the same cell culture induced by the same inducers.

While the ATRA→BDNF group exhibited typical neuronal-like phenotypes, these neuronal-like cells represented a small population among the larger population of elongated and other unchanged cells. Similarly, Kiraly et al. [40] reported functional typical neuronal-like cells differentiated from hDPSCs, however they are a small proportion compared with the entire cell culture and the authors interpreted that it may be because of the high proliferative capacity of the undifferentiated early hDPSC passages. This may be one explanation for the presence of small typical neuronal-like population in our study as we used early passage hDPSCs. Another explanation may be that there is a variation in the response of the hDPSCs to neurogenic induction due to the inherent heterogeneity of DPSCs [117]. While the highest TUBB3 expression was detected in the ATRA→BDNF group, this neuronal cytoskeleton marker [96] was also shown in control hDPSC cultures. TUBB3 is a cytoskeletal protein of neuronal cells and is required in neurodevelopment for guidance, differentiation, survival of neuronal cells [96,119] and axonal regeneration [120]. This marker has been considered as a

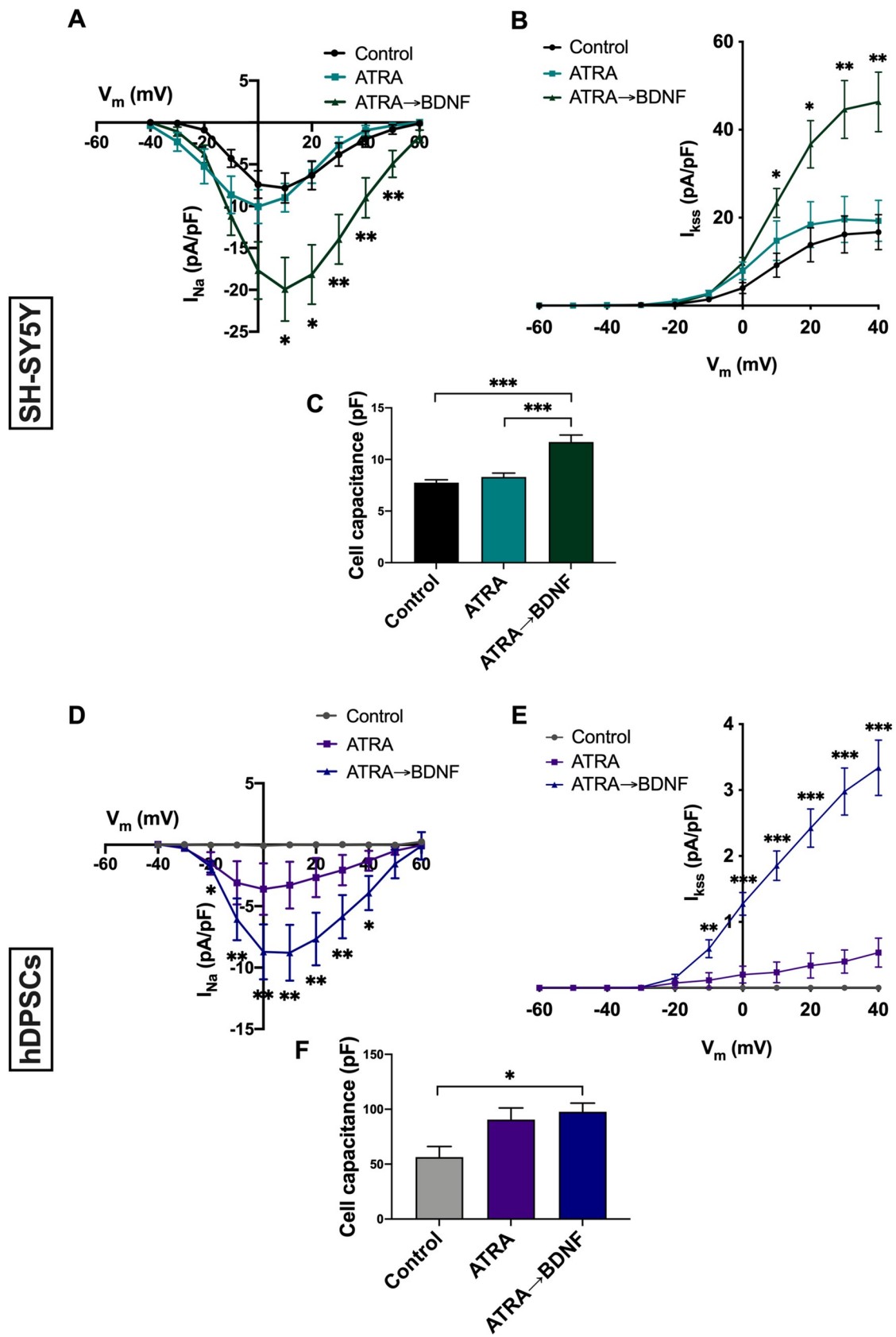

**Fig 5. Patch-clamp electrophysiology analysis of neuronal differentiated SH-SY5Y and hDPSCs.** (**A**) Mean I-V relationship of $I_{Na}$ in SH-SY5Y (control, n = 23; ATRA, n = 25; ATRA→BDNF, n = 21). (**B**) Mean I-V relationship for $I_{Kss}$ in SH-SY5Y (control, n = 22; ATRA, n = 18; ATRA→BDNF, n = 23). (**C**) Mean cell capacitance in SH-SY5Y (control, n = 45; ATRA, n = 43; ATRA→BDNF, n = 44). (**D**) Mean I-V relationship of $I_{Na}$ in hDPSCs (control, n = 6; ATRA, n = 10; ATRA→BDNF, n = 14). (**E**) Mean I-V relationship of $I_{Kss}$ in hDPSCs (control, n = 7; ATRA, n = 6; ATRA→BDNF, n = 10). (**F**) Mean cell capacitance in hDPSCs (control, n = 8; ATRA, n = 6; ATRA→BDNF, n = 23). Data were analyzed by Friedman repeated measure ANOVA for comparison between measured membrane potentials of each group and Kruskal-Wallis test with a pairwise comparison for comparison between experimental groups; significance values were adjusted by Bonferroni correction. Data plotted as mean ± SEM; *, ** and *** denote P< 0.05, P< 0.01, and P< 0.001.

specific marker for neuronal cells and widely used in neuronal differentiation studies. However, ours and other studies have detected TUBB3 expression in undifferentiated DPSCs [68,121]. TUBB3 has also been found in non-neuronal cells such as tumor cells and normal cells such as fibroblast, stroma cells, endocrine cells [122], and perivascular cells, including smooth muscle, and pericytes [123]. Thereby, the detection of the TUBB3 does not necessarily indicate neural differentiation, although its increase in expression most likely does. Consequently, use of multiple neuronal markers, particularly specific lineage ones, besides neuronal morphological change, and functional evidence are preferable to investigate and assess neuronal differentiation.

The ATRA→BDNF group demonstrated increased gene expression of neuronal markers compared with the ATRA alone exposure group. Furthermore, the gene expression of specific neuronal markers indicates that the differentiation was guided toward specialized neuronal lineage. For instance, the ATRA→BDNF group demonstrated gene upregulation of cholinergic neurotransmitter markers: choline acetyltransferase (CHAT) and acetylcholinesterase (ACHE) which are responsible for synthesis of neurotransmitter acetylcholine and modulation and termination of synaptic transmission function of neurotransmitter acetylcholine at postsynaptic cholinergic junction, respectively [124–126]. Simultaneously, the ATRA→BDNF group showed no change or absence of detection of the noradrenergic marker (DBH) [104] which is responsible for production of the norepinephrine neurotransmitter for synaptic transmission function of noradrenergic neurons. This upregulation of cholinergic markers combined with no change or absence of noradrenergic marker suggest a specialized cholinergic identity of the resultant neuronal-like cells derived from hDPSCs and SH-SY5Y cells. Establishment of cholinergic neuronal cells derived from hDPSCs are previously reported in different methodology studies [127–129]. It is also reported in other stem cell studies used ATRA and BDNF in combination with other inducers to differentiate stem cells into neuronal-like cells [130,131]. In addition, the cholinergic identity of the SH-SY5Y-derived neuronal-like cells induced by similar methodology was previously reported [73,74]. This consistency in the findings indicates that the ATRA and BDNF inducers are responsible for the cholinergic synaptic activity of the established neuronal-like cells.

Other specific markers were sensory neuronal marker (POU4F1/BRN3A) [103] and nociceptive voltage-gated sodium channel marker (SCN9A/Na$_v$ 1.7) [102] which were significantly expressed in the hDPSC ATRA→BDNF group. The POU4F1/BRN3A plays a neurodevelopmental role for sensory neurons [132] whereas SCN9A/Na$_v$ 1.7 is responsible for pain sensation detected by sensory neurons [133]. These expressions of the sensory neuronal markers in concomitant with significant reduced levels of motor neurodevelopmental marker (MNX1/HLXB9/HB9) [101] in the established hDPSC-derived neuronal-like cells indicate a specific guided differentiation toward the sensory neuronal lineage. Interestingly, these data are not entirely in agreement with previously mentioned studies [130,131] which used ATRA and BDNF with other combinations for stem-cell neuronal differentiation resulting in motor neuronal differentiation. The possible explanation for motor neuronal differentiation in these

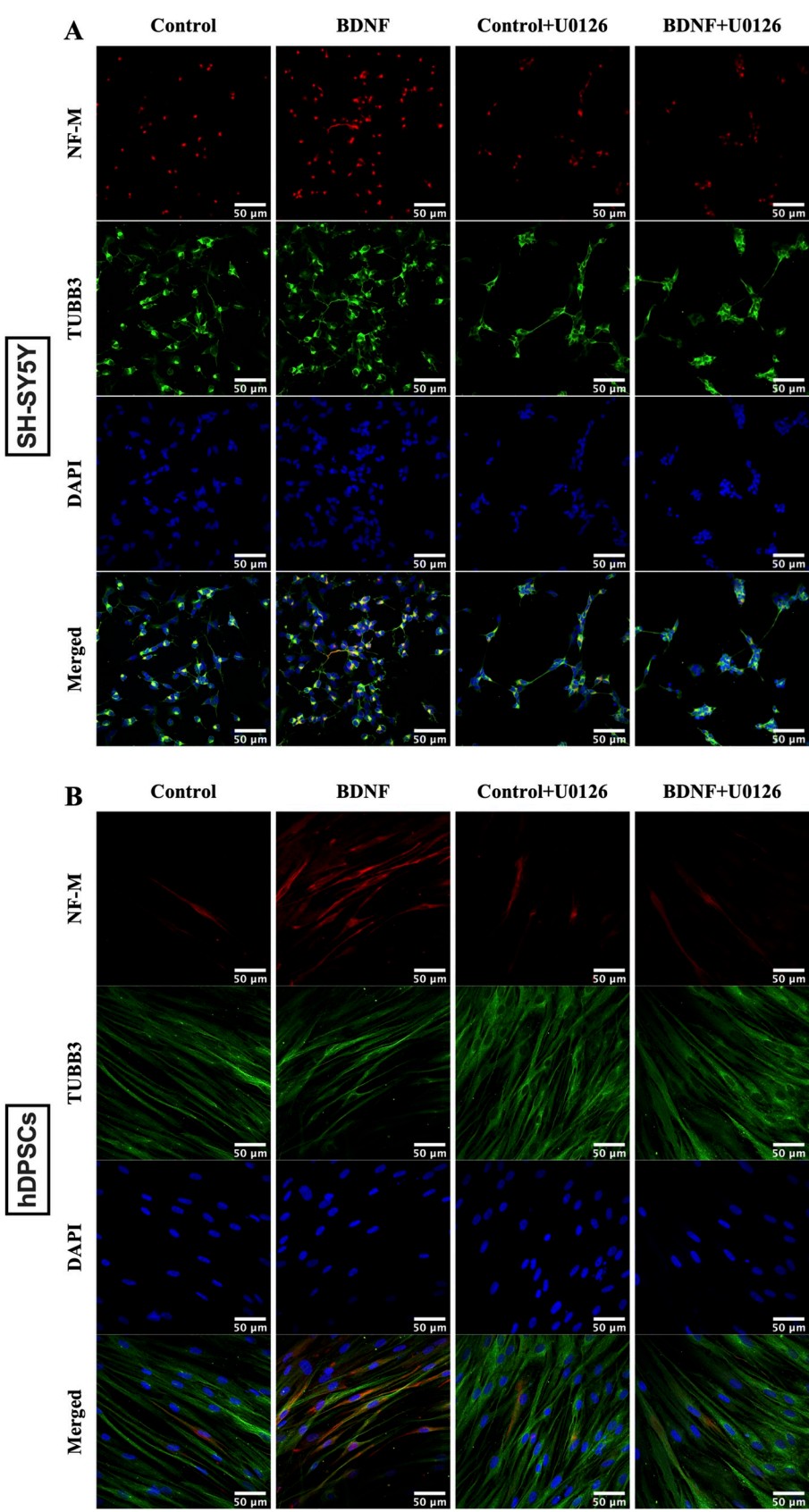

**Fig 6. Immunolabelling analysis of the mature marker (NF-M) expression in the presence and absence of the ERK1/2 inhibitor (U0126).** (A) SH-SY5Y cells. (B) hDPSCs. All groups were initially differentiated with ATRA for 5 days as a preparatory stage, followed by immunostaining and confocal microscopy analysis after 48h-incubation with BDNF in the presence or absence of the ERK/MEK inhibitor. Scale bar is 50 μm in all images.

studies is the presence of the sonic hedgehog (SHH) supplementation in the differentiating media which is reported as an active inducer for guided differentiation into the motor neuronal lineage [134,135]. In addition, the use of very low concentrations of ATRA (1nM to 2μM) and BDNF (10–20 ng/ml) in these studies may not be sufficient to guide the differentiation toward a sensory neuronal lineage in comparison with the current study (10μM ATRA and 50 ng/ml BDNF). In this context, there are some studies reported that disruption or lack of the BDNF disturbs the sensory neural development but not motor neural development [136–138] which indicate the BDNF supplementation in our study is the underlying inductive agent for the sensory identity of the established neuronal like cells. Therefore, the combinations and concentrations of multiple inducers should be carefully selected to guide the differentiation toward a specific neuronal lineage and avoid the possibility of non-specific or random neuronal differentiation.

It has previously been reported that the neuronal-like morphology and expression of certain neuronal markers in stem cells can result from chemical toxicity or stress in the differentiating media rather than being due to actual neuronal differentiation [139]. Hence, electrophysiology recordings were performed to investigate the functional electrical properties of the differentiated hDPSCs. The study data demonstrates that only the ATRA→BDNF group displayed

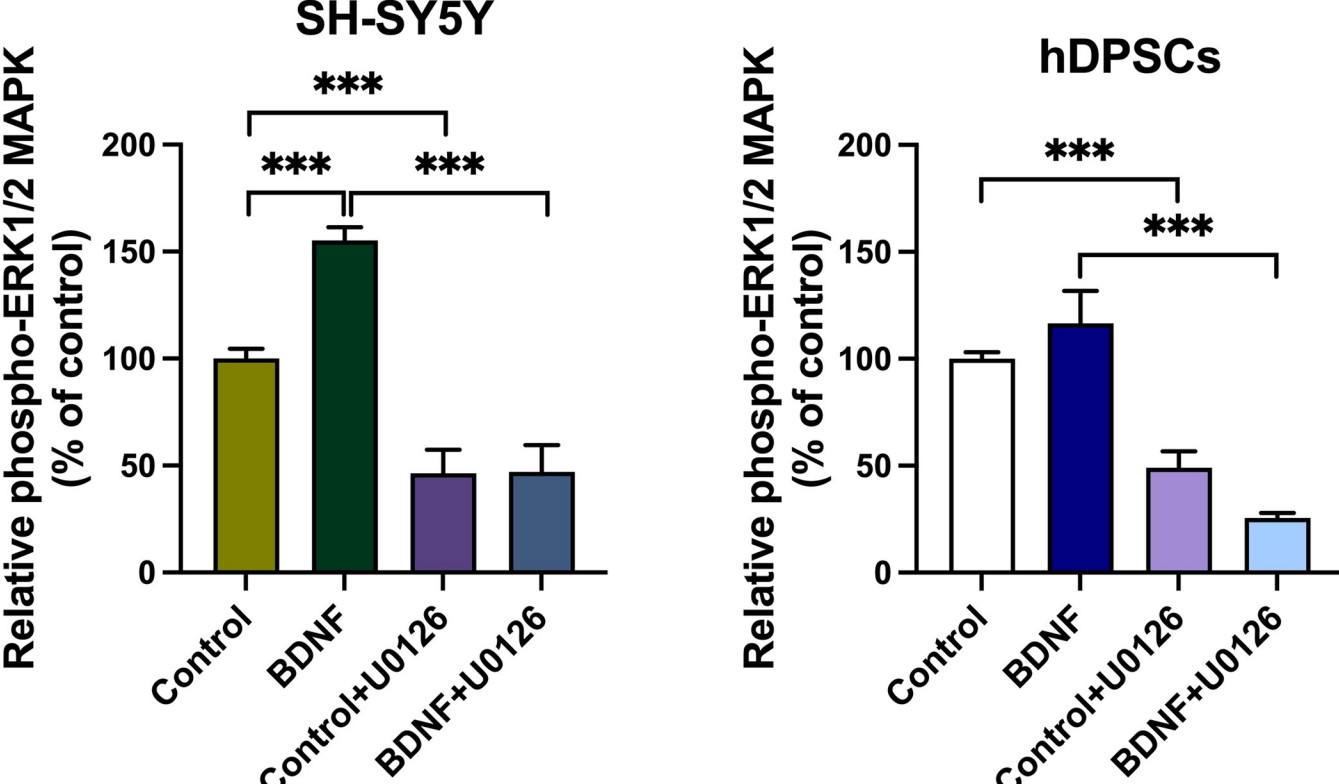

**Fig 7. Analysis of p44/42 MAPK (ERK1/2) phosphorylation in neuronal differentiated SH-SY5Y and hDPSCs in the presence or absence of the ERK/ MEK inhibitor (U0126).** These data were determined by quantitative sandwich ELISA and the absorbance values were read at 450nm. The data were analyzed by One-way ANOVA and Games-Howell Post-Hoc for pairwise comparison between the experimental groups. Data plotted as mean ± SD (n = 4; $^*$P< 0.05, $^{**}$P< 0.01, and $^{***}$P< 0.001).

significant $I_{Na}$ and $I_{Kss}$, whereas the ATRA group did not. This was consistent with the SH-SY5Y cell data and with Goldie et al. [74] who reported that ATRA alone produces intermediate differentiation and that addition of BDNF significantly induces the synaptic and functional transcriptional networks of SH-SY5Y cells. Similar data is also reported in other stemcell neuronal differentiation studies which used ATRA and BDNF in their inductive media [130,131]. Previous observations suggest that hDPSCs have a considerably larger cell capacitance compared to primary neurons and neuronal cell lines [117] and this was also apparent in the current study in which the measurements are similar to those reported previously [117]. As measurements were normalized to cell capacitance, this likely accounts for the smaller $I_{Kss}$ in the hDPSCs compared with SH-SY5Y cells. The smaller $I_{Kss}$ and $I_{Na}$ may also be a consequence of the immature electrophysiological phenotype in the hDPSCs which is a known limitation of many different types of human stem cell derived cell types [117,140]. Importantly, the current study indicate that additional BDNF supplementation in both SH-SY5Y and hDPSCs leads to significant induction of $I_{Na}$ and $I_{Kss}$, suggesting that this is an important factor for promoting 'neuronal-like' functional electrophysiological development.

In this study, the role of the ERK1/2 signaling in the neuronal differentiation of hDPSCs was investigated. The ERK/MEK inhibitor (U0126) significantly decreased the immunocytochemical expression of the NF-M marker and the EKR1/2 phosphorylation induced by BDNF supplementation. This outcome is consistent with findings of similar methodological study reported by Encinas et al. [75] who demonstrated that the BDNF induces the neuronal differentiation of ATRA-treated SH-SY5Y cells via ERK/MAPK signaling and ATRA pretreatment activates the Trk B receptor which subsequently increases BDNF binding and triggering of the ERK/MAPK signaling. In addition, the BDNF-mediated neurogenesis and neuritogenesis via ERK/MAPK signaling has been also reported in stem cells such as blood-derived mesenchymal stem cells [84], and immature progenitor neuronal cells [141–143]. Taken together, this provides convincing evidence to support the notion that BDNF induces neurogenesis of stem cells, including, hDPSCs in this study via ERK/MAPK signaling.

There was also reduction in phospho-ERK1/2 levels of the control group with the inhibitor applied, however no change in the immunocytochemical expression of NF-M neuronal marker was observed compared with controls in both cell types. This reduction in phospho-ERK1/2 levels was expected as the ERK signaling is central to the MAPK pathway which underpins a variety of biological processes [144,145]. Moreover, the immunocytochemical results showed no change in the expression of the NF-M mature neuronal marker, and this indicates that the reduction in ERK-phosphorylation of the control group when the inhibitor applied is not related to differentiation process. The present study suggests the involvement of ERK/MAPK pathway in both control and supplemented groups, but BDNF supplementation increased the triggering of this pathway to induce neuronal differentiation.

In conclusion, this study provides original evidence for differentiation of human DPSCs into neuronal-like cells, particularly toward cholinergic sensory neuronal cells. The hDPSC-derived cholinergic sensory neuronal-like cells may provide a suitable in *vitro* model to study neural function and nerve regeneration and could be harnessed for tissue-engineering constructs and regenerative transplantation therapies in medicine and dentistry. In addition, the combination of ATRA and BDNF may be an attractive therapy for neural regeneration, particularly for sensory cholinergic nerves.

## Supporting information

**S1 Table. hDPSCs information and stem cells characterization.**
(PDF)

**S2 Table. The antibodies and dilutions used for immunocytochemical analysis.**
(PDF)

**S3 Table. The primer details used for real-time PCR.**
(PDF)

**S1 File. Neurogenic differentiation protocol for neuroblastoma cell line (SH-SY5Y) and human dental pulp stem cells (hDPSCs).**
(PDF)

## Acknowledgments

The authors wish to thank our lab technicians at School of Dentistry, University of Birmingham namely Gay Smith, Michelle Holder, and Helen Wright for technical help and guidance during the laboratory work of this project. Special thanks to Dr Huzaimi Haron from School of Pharmacy, University of Birmingham for providing SH-SY5Y neuroblastoma cell line to be used in this project.

## Author Contributions

**Conceptualization:** Arwa A. Al-Maswary, A. Damien Walmsley, Paul R. Cooper, Ben A. Scheven.

**Data curation:** Arwa A. Al-Maswary, Molly O'Reilly, Andrew P. Holmes.

**Formal analysis:** Arwa A. Al-Maswary, Molly O'Reilly, Andrew P. Holmes.

**Funding acquisition:** Arwa A. Al-Maswary, A. Damien Walmsley, Paul R. Cooper, Ben A. Scheven.

**Investigation:** Arwa A. Al-Maswary, Molly O'Reilly, Andrew P. Holmes.

**Methodology:** Arwa A. Al-Maswary, Molly O'Reilly, Andrew P. Holmes, A. Damien Walmsley, Paul R. Cooper, Ben A. Scheven.

**Project administration:** A. Damien Walmsley, Paul R. Cooper, Ben A. Scheven.

**Supervision:** A. Damien Walmsley, Paul R. Cooper, Ben A. Scheven.

**Validation:** Arwa A. Al-Maswary, Molly O'Reilly, Andrew P. Holmes.

**Visualization:** Arwa A. Al-Maswary, Molly O'Reilly, Andrew P. Holmes.

**Writing – original draft:** Arwa A. Al-Maswary, Molly O'Reilly, Andrew P. Holmes.

**Writing – review & editing:** Arwa A. Al-Maswary, Molly O'Reilly, Andrew P. Holmes, A. Damien Walmsley, Paul R. Cooper, Ben A. Scheven.

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
