## [Decision Letter · Decision Letter 0]

18 May 2022

PONE-D-22-06526Exploring the neurogenic differentiation of human dental pulp stem cellsPLOS ONE

Dear Dr. Al-Maswary,

Thank you for submitting your manuscript to PLOS ONE. After careful consideration, we feel that it has merit but does not fully meet PLOS ONE’s publication criteria as it currently stands. Therefore, we invite you to submit a revised version of the manuscript that addresses the points raised during the review process.

We look forward to receiving your revised manuscript.

Kind regards,

Sujeong Jang

Academic Editor

PLOS ONE

Journal Requirements:

Reviewers' comments:

Reviewer's Responses to Questions

**Comments to the Author**

1. Is the manuscript technically sound, and do the data support the conclusions?

Reviewer #1: Yes

Reviewer #2: Partly

2. Has the statistical analysis been performed appropriately and rigorously? 

Reviewer #1: Yes

Reviewer #2: Yes

3. Have the authors made all data underlying the findings in their manuscript fully available?

Reviewer #1: Yes

Reviewer #2: Yes

4. Is the manuscript presented in an intelligible fashion and written in standard English?

Reviewer #1: Yes

Reviewer #2: No

5. Review Comments to the Author

Reviewer #1: In this article, the authors cultured hDPSCs with a novel treatment with ATRA and BDNF sequentially. Compared to the SH-SY5Y neurogenic differentiation model, hDPSCs can be differentiated to a cholinergic sensory neuronal lineage with the above treatment. Based on this result, the authors connected the ERK1/2 phosphorylation with BDNF treatment during hPDSCs neurogenic differentiation. The experimental design is reasonable and systematic. The some major concerns should be addressed before accept to publication.

1.The result of immunocytochemistry is semi-quantitative, authors should provide the result of western blot of NF-M, TUBB3, and GFAP.

2.hDPSCs can secret levels of BDNF when cultured in vitro. Have you detected whether BDNF was secreted before you added BDNF? Did BDNF secreted by hDPSCs had an effect on their neurogenic differentiation?

3.Authors only detected the expression of NF-M and TUBB3 after using U0126, which suggested U0126 impeded neurogenic differentiation. The evidence for the conclusion that ERK/MAPK-mediated sensory cholinergic neuronal differentiation of hDPSCs is not enough.

4.Figure 6, what is the meaning of the comparison between Control+U0126 and BDNF, and between Control and BDNF+U0126? I prefer to show the result in the order of Control, BDNF, Control+U0126, and BDNF+U0126. And the result that relative phosphor-p44/42 MAPK in BDNF+U0126 group is lower than that in control+U0126 group is not reasonable. How to explain it?

5.The sentence “These results suggest the involvement of ERK/MAPK pathway in control and supplemented groups…” is ambiguous.

6.There is some researches about hDPSCs neurogenic differentiation with the treatment of BDNF synergistically (Goudarzi G, et al. 2020; Gonmanee T, et al. 2018). It is easy to predict that BDNF can promote hDPSCs neurogenic differentiation.

Reviewer #2: Intro

1. First statement is too generic. Please introduce WHY stem cells have received such “special attention”.

2. The authors need to introduce the indications/advantages of stem cell transplantation before mentioning the disadvantages. The reader is left alone to understand the reasons.

3. Discuss WHY reedy in vitro is promising. Again, the reader is left alone to understand such excitement

4. Neuronal cell models derived… Why they are useful? Which problem in the neuroscience field they can solve? So far, the introduction has too many “great applications” without any context.

5. DPSC introduction is too short and (again) only contains advantages. Please discuss the WHYs. Introduce why DPSC has potential in the lights of their origin, markers and intrinsic “neurogenic potential” in the lights of relevant papers such as (but not limited to)

- 10.1634/stemcells.2007-0979

- 10.1016/j.archoralbio.2019.104572

- 10.1186/scrt419

- 10.1155/2013/250740

6. Second para fails to establish the importance of the protocol used. The authors need to discuss the advantages and disadvantages of the many protocols available. So far, the text reads like “there are many protocols, we selected one”. The authors need to discuss that some protocols take too long, are too expensive, too laborious. However, some published “straightforward” protocols have never been validated, lack functional assays, have never tested by more than a group. Since the authors are putting attention to a “relatively straightforward 2-component method”, the reader needs to be educated about the pros/cons of current protocols. Please improve your introduction discussing the findings from previous papers to allow the reader why have you selected such “straightforward 2-component method”. Suggested literature (many other papers have reported differentiation protocols for DPSC):

- 10.1634/stemcells.2007-0979

- 10.1016/j.archoralbio.2019.104572

- 10.1080/00207454.2019.1664518

- 10.1155/2013/250740

- 10.1186/scrt419

7. are reportedly the main pathways within the nervous systems: please check/rephrase this para. main pathways? What do the authors mean about it? The nervous system is very complex and include sensory, motorneurons… there are many “main” pathways depending on the type of neurons.

8. Methods: please describe the protocol used to promote the neurogenic differentiation in full. Unfortunately, the authors do not provide the methods used and Fig 1 does not provide clear instructions on how the differentiation was performed neither what are indeed the groups. It is very confusing. This is a major concern about this work since the authors claim that this was as “simple method” but in reality, it is not possible to understand how this work was done. The authors suggest that the reader search for the paper published by Encinas but in reality, the protocol use for this study should be available. It is not possible to understand the methods and Fig 1 fails to show how the study was done or how the groups were organized. Please describe the methods that are the base of your study in full.

9. Please remove “To the best of our knowledge” and rephrase that statement. At present, the reviewer cannot agree that this is a “relatively simple approach” since the methods are not well described. Also, please remove the word “relatively” from the manuscript. Relatively to what? Also, “relatively” may be denote a personal opinion. Summary: please keep the scientific tone and forego “to our knowledge, relatively, simplified….” Because the context of other protocols is not presented yet.

10. by the ATRA�BDNF protocol: one cannot understand what do the authors mean by the ATRA�BDNF protocol.

11. Please state in the methods/results/discussion the specific reasons for using ATRA and BDNF in this work. What are the reasons and expected outcomes? “Atra was used to induce a b and c as reported by xyz…”

12. Apparent neuronal morphological features: please describe the features.

13. Going back the results the reviewer is confronted by “BDNF and FBS supplementation (ATRA�0% serum)” but again, the methods and fig 1 do not show evidently how the work was performed. It becomes very difficult to review the work as it is up to the reviewer/reader to understands the arrangement and rationale for the controls, additional groups… The authors need to describe the work in full. Also, include description (in methods, results or discussion) for some of the choices. For instance “we added the group ATRA�0% serum to evaluate this and that” or “the group ATRA�0% serum was added to control the presence of something…” This seems to be a well executed work, but it is not possible to understand the procedures and rationale for many of the groups tested/information presented. It may make sense for the authors, but for those reading the work for the first time, it is nearly impossible to understand rationales/procedures.

14. heterogeneity of specific markers in hDPSC cultures which results in the guiding of the cells towards specific neuronal lineages: please explain. What do the authors mean about heterogeneity of specific markers and which are the “specific lineages”. Please note that it is up to the reader to read papers 53-54 to understand this statement.

15. relatively small population: remove the word “relatively” from the manuscript and use objective terms.

16. tested hDPSCs were characterized as stem cells, however, they had not been cell-sorted resulting in low percentage of stem cells in these DPSC cultures: this seems to be a too convenient explanation. If the authors used DPSC (although not all are DPSC), then blaming on cell sorting is very strange. Either the authors used DPSC or not. If cell isolation and characterization were enough to confirm these were dpsc (and the authors used them!)… it is not acceptable to read that, maybe, only a small percentage was dpsc. If this is the case, the basis of this work is compromised and this is not a work on neurogenic differentiion of DPSC. Please explain this contradiction.

17. The TUBB3 expression has previously been reported in undifferentiated stem cells, including DPSCs. Please explain the relevancy of such marker. Just mentioning that it has been previously observed does not add much value to the discussion.

18. Please explain the most relevant markers observed in your work. Do note that the reader may not understand fully the relevancy of markers such as Tub3, Chat and others. Hence, discuss your findings and the the roles of ACHE and CHAT on DPSC neurogenic differentiation/functions in the lights of

- 10.1016/j.jfma.2014.09.003

- 10.22203/eCM.v041a16

- 10.1002/jcp.24570

- 10.1002/jcp.25314

19. Please consider having easier names to your groups… what shall the reviewer understand about hDPSC ATRA�BDNF group. Is this the name of the group? Does it mean that hDPSC treated with ATRA and subsequently with BDNF? This goes back to the comments about the methods, it is so unclear that it becomes very difficult to even assess the findings. This is a major concern for this paper. Do not that so far very few comments are given to the results because it is nearly impossible to understand how this work was performed (hence the data and controls cannot be even assessed at this point).

20. The possible explanation for this is that the sonic hedgehog: this statement seems to be incomplete.

21. addition, their use: should be the use

22. into sensory cholinergic neuronal-like cells: this seems to be an overestimation of the results. Have the authors measured choline release? The reviewer understand that the markers are present, but this is a hallmark of choline production and release. Please rephrase or highlight the evidence that confirms that these are functional cholinergic neurons.

Overall, this paper has a lot of potential, but the writing needs to be much improved. The reader is left alone to understand the procedures and relevancy of the work. At this stage, it is not possible to discuss the data in depth because it is not clear how the work was done, which controls were employed (it is true that the authors mention, but the reader shall not make all the effort to understand how this work was done). Hence, the reviewer cannot analyze the data without knowing for sure how that work was done and which group served as a control for what… The authors are invited to submit a revised version that will allow the reviewer to be sure that everything is on place. It is a nice paper but needs to be more self-contained.

6. PLOS authors have the option to publish the peer review history of their article (what does this mean?). If published, this will include your full peer review and any attached files.

Reviewer #1: No

Reviewer #2: No

---

## [Author Response · Author response to Decision Letter 0]

25 Jul 2022

Point-by-point response to editor’s and reviewers’ comments

Reference: PONE-D-22-06526 

Exploring the neurogenic differentiation of human dental pulp stem cells

PLOS ONE

We would like to thank the editor “Prof. Sujeong Jang” and reviewers for their time and efforts to review our manuscript with reference: PONE-D-22-06526, title: Exploring the neurogenic differentiation of human dental pulp stem cells. We found the comments constructive and have updated our manuscript accordingly.

We have responded to each point (see below: dark blue text). 

Note: All page numbers and lines refer to the revised manuscript with track changes “the word file” as the text moved in the built pdf.

Thank you again for your thoughtful commentary!

Sincerely,

Authors: Arwa A. Al-Maswary, Molly O'Reilly, Andrew P. Holmes, A. Damien Walmsley, 

Paul R. Cooper, and Ben A. Scheven.

5. Review Comments to the Authors

Reviewer #1: In this article, the authors cultured hDPSCs with a novel treatment with ATRA and BDNF sequentially. Compared to the SH-SY5Y neurogenic differentiation model, hDPSCs can be differentiated to a cholinergic sensory neuronal lineage with the above treatment. Based on this result, the authors connected the ERK1/2 phosphorylation with BDNF treatment during hPDSCs neurogenic differentiation. The experimental design is reasonable and systematic. The some major concerns should be addressed before accept to publication.

Comment 1: The result of immunocytochemistry is semi-quantitative, authors should provide the result of western blot of NF-M, TUBB3, and GFAP.

Response: 

We agree with the reviewer that immunocytochemistry is not a quantitative assay, but the immunocytochemical staining is still a valid approach to show the differences, albeit qualitative, in the expression of the neuronal markers “NF-M, TUBB3, and GFAP” between the experimental groups as highlighted in the manuscript. To strengthen the immunocytochemical observations, our study used the quantitative “real-time RT-PCR” to quantify the changes in gene levels of the neuronal markers in the experimental groups.

Comment 2: hDPSCs can secret levels of BDNF when cultured in vitro. Have you detected whether BDNF was secreted before you added BDNF? Did BDNF secreted by hDPSCs had an effect on their neurogenic differentiation?

Response: 

We thank the reviewer for raising this point. In this study, we have not analysed the secretion of BNDF by hDPSCs and whether this had an additional effect on cell differentiation. Although DPSCs have been shown to secrete BDNF in our previous work, we believe it is unlikely that this had any significant influence to induce neurogenic differentiation for the following reasons:

1. No neurogenic changes were observed in the 2nd parallel control group which is exposure to ATRA followed by serum-free media “ATRA→0 serum” compared with ATR→ABDNF group. This group is identical to successful differentiated group “ATRA→BDNF” but without BDNF supplementation. If there is an effect, it will be clear in this control culture. On the other hand, the significant changes were clearly detected when the culture was supplemented by BDNF. 

2. The DPSCs secrete very low levels of BDNF as reported in the literature: 85.5 pg/ml produced from 70–80% cell confluency in the T75 flask incubated for 72 h (sultan et al., 2020 and 2021), 1600.6 ± 338 pg for 100,000 cell seeding incubated for 48 h (Mead et al., 2013), ~1000 ± 1000 pg/ml for 10,000 cell seeding incubated for 14 days whereas the stimulated DPSCs (were induced by growth factors for 14 days to increase the secretome output) produce 4000 ± 2000 pg/ml of the BDNF (Kolar et al., 2017). These reported amounts are much lower than our BDNF supplementation (50 ng/ml= 50,000 pg/ml) which was supplemented into the culture at least 2 times (100 ng/ml =100,000 pg/ml) during BDNF supplementation period (7 days). 

References:

Sultan et al., 2020 DOI: 10.1038/s41598-020-76684-0

Sultan et al., 2021 DOI: 10.4103/1673-5374.306089

Mead et al., 2013 DOI: 10.1167/iovs.13-13045

Kolar et al., 2017 DOI: 10.1038/s41598-017-12969-1

3. We indirectly discussed this point in the introduction section “in differentiating methods’ paragraph” as one study differentiated mice DPSCs into neuronal-like cells by culturing in only serum-free media and it is thought that the neurotrophic factors released from the cells are the inducers. Our discussion for this point is as following (highlighted with yellow background in the manuscript, page 5, line 8 to 22):

“On the other hand, one protocol reported that serum-free media without any supplementations can differentiate mice DPSCs into neuronal-like cells and expressed MAP2, nestin, and Tub3/βIII-tubulin neuronal markers [67]. However, these neuronal markers have been reported in non-differentiated DPSCs [41, 68] which may not provide sufficient evidence for neuronal differentiation, particularly with no functional testing. Furthermore, this serum-free protocol has been recently used by Madanagopal et al., [69] as one of three protocols to differentiate hDPSCs into neuronal cell type. This study reported that the serum-free media alone did not result in neuronal differentiation of hDPSCs compared with what has been reported in mice DPSCs by Zainal et al., [67] and the authors interpreted that this may occur due to the genetic and physiological differences between mice and human [69, 70]. In addition, the concept of neuronal differentiation in serum-free media without supplementations is previously discussed by Croft and Przyborski [71] who reported that culturing in serum-free media is an environmental culture stressor which results in pseudo-neuronal morphology and expression “artifacts”.”

Therefore, BDNF may be released from DPSCs during the neurogenic differentiation which might support the differentiation, however the concentration would not be sufficient to solely induce the neurogenic differentiation of DPSCs without the exogenous BDNF supplementation.

Comment 3: Authors only detected the expression of NF-M and TUBB3 after using U0126, which suggested U0126 impeded neurogenic differentiation. The evidence for the conclusion that ERK/MAPK-mediated sensory cholinergic neuronal differentiation of hDPSCs is not enough.

Response: 

We appreciate the reviewer’s comment. However, we believe that our work using the immunostaining of a specific mature neuronal marker (NF-M) linked with quantification of phospho-ERK proteins provides a plausible indicator at the least for ERK/MAPK involvement in this neuronal differentiation. In addition, our results are supported by previously published “similar methodology” research in SH-SY5Y cells (Encinas et al., 1999) that also demonstrated the involvement of ERK/MAPK in neurogenic differentiation. Moreover, the BDNF-mediated neurogenesis and neuritogenesis via ERK/MAPK is reported in other cells such as blood-derived mesenchymal stem cells (Lim et al., 2008), and immature progenitor neuronal cells (Alonso et al., 2004; Ortega and Alcántara, 2009; Lee et al., 2016). Taken together, this provides convincing evidence to support the notion that BDNF induces neurogenesis of stem cells, including, hDPSCs via ERK/MAPK signaling. In the revised manuscript, we have now further discussed this point by adding these studies to the manuscript (highlighted with yellow background in the manuscript, page 25, line 23 to page 26, line 2).

“In addition, the BDNF-mediated neurogenesis and neuritogenesis via ERK/MAPK signaling has been also reported in stem cells such as blood-derived mesenchymal stem cells [84], and immature progenitor neuronal cells [141-143]. Taken together, this provides convincing evidence to support the notion that BDNF induces neurogenesis of stem cells, including, hDPSCs in this study via ERK/MAPK signaling.”

References:

Encinas et al., 1999 DOI: 10.1046/j.1471-4159.1999.0731409.x

Lim et al., 2008 DOI: 10.1002/jnr.21669

Alonso et al., 2004 DOI: 10.1101/lm.67804

Ortega and Alcántara, 2009 DOI: 10.1093/cercor/bhp275

Lee et al., 2016 DOI: 10.1016/j.brainresbull.2016.01.002

 

Comment 4: Figure 6, what is the meaning of the comparison between Control+U0126 and BDNF, and between Control and BDNF+U0126? I prefer to show the result in the order of Control, BDNF, Control+U0126, and BDNF+U0126. And the result that relative phosphor-p44/42 MAPK in BDNF+U0126 group is lower than that in control+U0126 group is not reasonable. How to explain it?

Response: 

“Figure 6, what is the meaning of the comparison between Control+U0126 and BDNF, and between Control and BDNF+U0126?”

The U0126 inhibitor greatly reduced the phosphorylation of the experimental groups. This consequently made each inhibited group (Control+U0126 or BDNF+U0126) significantly statistically lower than the other “unpaired” non-inhibited group (BDNF and Control), particularly Control versus BDNF+U0126. However, these “secondary results” may confuse and distract the reader from the main findings. Thus, the only the important and significant aspects of the data are now highlighted in the graph and written results’ section (see figure 7: It is now figure 7 not figure 6 as we rearranged and modified the section “ERK1/2 inhibitor blocked neuronal differentiation and ERK1/2 phosphorylation” to clarify the results). 

“I prefer to show the result in the order of Control, BDNF, Control+U0126, and BDNF+U0126”.

Thank you for this suggestion. The groups of the ELISA and immunocytochemistry results were reordered according to your suggestion “Control, BDNF, Control+U0126, and BDNF+U0126”. Now, they look much better and clearer results.

“And the result that relative phospho-p44/42 MAPK in BDNF+U0126 group is lower than that in control+U0126 group is not reasonable. How to explain it?”

Yes, we agree that is not logical and unexpected that the phosphorylation of control+U0126 group is higher than that of BDNF+U0126 group in hDPSCs. At this stage, we cannot give a sound explanation based on our data, and this might be due to an “as yet” unknown reason or process. 

Comment 5: The sentence “These results suggest the involvement of ERK/MAPK pathway in control and supplemented groups…” is ambiguous.

Response: 

Thank you for this comment. The sentence has been clarified as the following (yellow-highlighted in the manuscript, page 19, line 20 to 25):

“the increase of the phospho-ERK1/2 levels in the BDNF-supplemented group compared with those of control group and concomitant reduction of the phospho-ERK1/2 levels in the pre-treated groups with ERK/MEK inhibitor (control+U0126 and BDNF+U0126) suggest the involvement of ERK/MAPK pathway in control and supplemented groups, however BDNF supplementation induced further activation of this pathway.”

In addition, the whole section of “ERK1/2 inhibitor blocked neuronal differentiation and ERK1/2 phosphorylation” has been modified and rearranged to make it clearer as shown in the track changes, pages 18 to 20.

 

Comment 6: There is some researches about hDPSCs neurogenic differentiation with the treatment of BDNF synergistically (Goudarzi G, et al. 2020; Gonmanee T, et al. 2018). It is easy to predict that BDNF can promote hDPSCs neurogenic differentiation.

Response: 

BDNF was one component of several supplementations in the differentiation protocols of both highlighted studies (Goudarzi G, et al. 2020; Gonmanee T, et al. 2018) that induced neuronal differentiation of the hDPSCs. However, it should be noted that these studies did not indicate BDNF as the principal supplementation for neurogenesis. It may be predicted that BDNF has neurogenic influence in those mixtures plus their physiological neuronal function as one of neurotrophins responsible for differentiation, survival, and maintenance of the nervous system in the human body (Huang and Reichardt, 2001; Ivanisevic and Saragovi, 2013). However, our study aimed to obtain evidence and found that BDNF uses as a sole supplement directly promoted neurogenic differentiation after pre-induction with ATRA supplementation and that in particular involved the specific neurogenic induction along the “cholinergic sensory lineage”.

Goldie et al., 2014 and Encinas et al., 2000 highlighted that ATRA supplementation for 5 days induces the primary neuronal differentiation and also activates the BDNF receptor (Trk-B receptor) on the cell membrane which maximize the BNDF binding and subsequently further potentiate the neuronal differentiation and maturation of SH-SY5Y cells. Thus, it was of interest whether this sequential treatment “ATRA→BDNF” would be effective on hDPSC neuronal differentiation and maturation, as was explored and addressed in this study. 

References:

Goudarzi G, et al. 2020 DOI: 10.5115/acb.19.241

Gonmanee T, et al. 2018 DOI: 10.1016/j.archoralbio.2018.01.011

Huang and Reichardt, 2001 DOI: 10.1146/annurev.neuro.24.1.677 

Ivanisevic and Saragovi, 2013 (https://doi.org/10.1016/B978-0-12-385095-9.00224-4).

Goldie et al., 2014 DOI: 10.3389/fncel.2014.00325

Encinas et al., 2000 DOI: 10.1046/j.1471-4159.2000.0750991.x

Reviewer #2: Intro

 

Comment 1: First statement is too generic. Please introduce WHY stem cells have received such “special attention”.

Response: 

Thank you for this comment. We have now added sentences to clarify why stem cells gained special attention for nerve regeneration as following (highlighted with a yellow background in the manuscript, see page 3, line 1-6):

“Over recent decades, stem cells have gained special attention for potential nerve regeneration to treat nerve injuries or defects [1, 2]. Clinical evidence suggests that current therapies offer limited functional nerve recovery [3] and there are other drawbacks with graft procedures such as nerve sacrifice and nerve mismatch [1]. In dentistry, for example in regenerative endodontics, there is a need for nerve regeneration to achieve functional pulp regeneration [4, 5] which regulates pulpal blood flow, defense, and reparative process [6, 7].”

Comment 2: The authors need to introduce the indications/advantages of stem cell transplantation before mentioning the disadvantages. The reader is left alone to understand the reasons.

Response: 

We supported this part in the introduction by adding additional sentences at the beginning to connect this paragraph with the previous one and then highlighting positive therapeutic outcomes of stem cell transplantation/therapy as follows (highlighted with yellow background in the manuscript, see page 3, line 7 to 15):

“Stem cells have the potential to differentiate into multiple cell types with neural stem cells giving rise to neuronal cells and their supporting cells “glial and Schwann cells” [8, 9]. As a result, the stem cells have been promoted as neuronal cell replacements for nerve repair and regeneration [10, 11]. These stem cells can be either transplanted alone [12] or as part of designed engineered tissue/conduit to replace the defective neuronal tissue [13, 14]. Stem cell transplantations have demonstrated positive therapeutic nerve regeneration, functional recovery, and neuronal survival in several neurological traumas such as brain injury [15] and spinal cord injury/transection [16, 17], optic nerve crush [18], and injured peripheral nerves [12].”

Comment 3: Discuss WHY ready in vitro is promising. Again, the reader is left alone to understand such excitement.

Response: 

We have added sentences to clarify why using ready in vitro differentiated cells derived from stem cells is promising as follows (highlighted with yellow background in the manuscript, see page 3, line 18 to page 4, line 1):

“For example, some studies demonstrated that transplantation of the predifferentiated stem cells into a neuronal phenotype “neuronal cell models” results in greater restoration of neuronal loss [21], enhances nerve regeneration and functional recovery in brain [21, 23], spinal cord [24], and peripheral nerve injuries [25, 26]. Interestingly, it has also been reported that neuronally differentiated stem cells secrete greater amounts of neurotrophic factors [25, 27]. Hence, these neuronal cell models are not only neuronal cell replacements for the neuronal injury or defect, but they may further boost the nerve regeneration via their neurotrophic secretions.”

Comment 4: Neuronal cell models derived… Why they are useful? Which problem in the neuroscience field they can solve? So far, the introduction has too many “great applications” without any context.

Response: 

We have added sentences to clarify why using “neuronal cell models derived from stem cells are also useful for in vitro studies in neuroscience” as following (highlighted with yellow background in the manuscript, see page 4, line 3 to 7):

“For example, these neuronal cell models can be used to study neurodegenerative diseases such as Parkinson’s disease [30] and Alzheimer [31], pharmacological-related topics “drug discovery and toxicity testing” [32, 33], and neurodevelopment and injury [28]. As these neuronal cell models are differentiated from primary stem cells, they are more appropriate for simulation of the physiological properties of in vivo neurons [34, 35].”

Comment 5: DPSC introduction is too short and (again) only contains advantages. Please discuss the WHYs. Introduce why DPSC has potential in the lights of their origin, markers and intrinsic “neurogenic potential” in the lights of relevant papers such as (but not limited to) - 10.1634/stemcells.2007-0979 - 10.1016/j.archoralbio.2019.104572 - 10.1186/scrt419 - 10.1155/2013/250740

Response: 

Thank you for this comment and the quoted articles. We have modified the paper accordingly with a separate paragraph covering DPSCs regarding their potential, origin, markers and intrinsic factors in the light of relevant papers…” as follows (highlighted with yellow background in the manuscript, see page 4, line 8 to page 5, line 4):

“Dental pulp stem cells (DPSCs) are primary ecto-mesenchymal stem cells (MSCs) that have gained attention as a potential source for neuronal regenerative therapies. The neurogenic potential of DPSCs is closely related to their embryonic origin and various biological characteristics. DPSCs are derived from cranial neural crest cells during tooth development [36, 37]. In this context, it has been demonstrated that DPSCs retain the properties of neural crest cells such as EphB/Ephrin-B molecules and Wnt1-marker in in vitro cell culture which possess the differentiation capacity into any neural crest-derived tissue, including neuron [38, 39]. In addition to the stem cell markers, the expression of the neural markers in non-differentiated DPSCs, such as musashi12, nestin, MAP2ab, βIII-tubulin, N-tubulin, and neurogenin-2 underline their potential for neuronal differentiation [40, 41]. Furthermore, DPSCs express neurotrophic factors such as NGF, GDNF, BDNF, and NT-3 which are demonstrated to have neurogenic and cell survival effects [42, 43]. Moreover, DPSCs have been considered to be able to differentiate into specific neuronal cells of nervous system depending upon the induced environment [44, 45] which make DPSCs an attractive cell source for specific neuronal-lineage regeneration therapies. Additionally, DPSCs exhibit other favorable non-neurogenic factors such as their unique immunomodulation properties which prevent the possibility of immune rejection/reactions [46, 47] or tumor formation [48, 49] which is reported in other stem cell transplantations [50, 51]. Finally, DPSCs are easily obtainable from teeth extracted for various dental reasons without raising ethical concerns [52]. The neuro-regenerative potential of hDPSCs have been highlighted for dental pulp regeneration [53, 54], retinal [55, 56] and nerve injury [57, 58]. Thus, DPSCs potentially offer a safe neurogenic-potential stem population suitable for multiple clinical neuronal therapeutic applications.”

 

Comment 6: Second para fails to establish the importance of the protocol used. The authors need to discuss the advantages and disadvantages of the many protocols available. So far, the text reads like “there are many protocols, we selected one”. The authors need to discuss that some protocols take too long, are too expensive, too laborious. However, some published “straightforward” protocols have never been validated, lack functional assays, have never tested by more than a group. Since the authors are putting attention to a “relatively straightforward 2-component method”, the reader needs to be educated about the pros/cons of current protocols. Please improve your introduction discussing the findings from previous papers to allow the reader why have you selected such “straightforward 2-component method”. Suggested literature (many other papers have reported differentiation protocols for DPSC): - 10.1634/stemcells.2007-0979 - 10.1016/j.archoralbio.2019.104572 - 10.1080/00207454.2019.1664518 - 10.1155/2013/250740 - 10.1186/scrt419

Response: 

Thank you for this point, the ideas, and references. The differentiation protocol paragraph has now been rewritten to discuss the other differentiating protocols and highlight why we chose the 2-component method in light of your suggested ideas, and reports…” as follows (highlighted with yellow background, see page 5, line 6 to 25):

“Most differentiating protocols for hDPSCs use complex mixture of supplements either in multiple stages [40, 45, 59, 60] or/and long culture duration “more than a month” [61-66] which make the procedures relative expensive and time-consuming. On the other hand, one protocol reported that serum-free media without any supplementations can differentiate mice DPSCs into neuronal-like cells and expressed MAP2, nestin, and Tub3/βIII-tubulin neuronal markers [67]. However, these neuronal markers have been reported in non-differentiated DPSCs [41, 68] which may not provide sufficient evidence for neuronal differentiation, particularly with no functional testing. Furthermore, this serum-free protocol has been recently used by Madanagopal et al., [69] as one of three protocols to differentiate hDPSCs into neuronal cell type. This study reported that the serum-free media alone did not result in neuronal differentiation of hDPSCs compared with what has been reported in mice DPSCs by Zainal et al., [67] and the authors interpreted that this may occur due to the genetic and physiological differences between mice and human [69, 70]. In addition, the concept of neuronal differentiation in serum-free media without supplementations is previously discussed by Croft and Przyborski [71] who reported that culturing in serum-free media is an environmental culture stressor which results in pseudo-neuronal morphology and expression “artifacts”. Moreover, there is little convincing evidence for successful functional DPSC neurogenic differentiation [61, 62]. Hence, there is a need for simple, and relatively rapid differentiating protocol underpinned with sufficient evidence for neuronal differentiation and functionality.”

Comment 7: are reportedly the main pathways within the nervous systems: please check/rephrase this para. main pathways? What do the authors mean about it? The nervous system is very complex and include sensory, motorneurons… there are many “main” pathways depending on the type of neurons.

Response: 

Thank you for this comment. What we intended to highlight was that these pathways have been reported in regulating the neuronal differentiation and survival. Consequently, this sentence is now rephrased as below (highlighted with yellow background in the manuscript, see page 6, line 12 to 15):

“Although, there are many signaling pathways active in the nervous system, mitogen-activated protein kinase (MAPK) and phosphatidylinositol-3-Kinase/protein kinase B (PI3K/Akt) are reportedly central and essential in the overall regulation of neural differentiation and survival [83, 84].”

Comment 8: Methods: please describe the protocol used to promote the neurogenic differentiation in full. Unfortunately, the authors do not provide the methods used and Fig 1 does not provide clear instructions on how the differentiation was performed neither what are indeed the groups. It is very confusing. This is a major concern about this work since the authors claim that this was as “simple method” but in reality, it is not possible to understand how this work was done. The authors suggest that the reader search for the paper published by Encinas but in reality, the protocol use for this study should be available. It is not possible to understand the methods and Fig 1 fails to show how the study was done or how the groups were organized. Please describe the methods that are the base of your study in full.

Response: 

We apologise that the protocol that we described was considered not in adequate detail. We have now provided further details and explained the steps of neurogenic differentiation of hDPSCs as follows (see below paragraph and table; highlighted with yellow background in the manuscript, pages 8 and 9). In addition, we explained in full the culturing, supplements’ preparation and differentiating steps in the supplementary S2 file (additional file).

“The seeded cells were incubated overnight to allow cells to attach to the culturing surface before conducting the differentiation experiment. Subsequently, the differentiation and control media were freshly prepared for each experimental group as described in Table 1 (all supplements were defrosted and immediately used for the experiment to avoid the material degradation over the time). After that, the overnight media were replaced with ATRA-supplemented (R 2625, Sigma-Aldrich, UK) or with control media and then incubated in a humidified incubator at 37 °C and 5% CO2. This media change step was performed in limited light in the laboratory room and the culture hood’s light was switched off due to the light-sensitive nature of ATRA. Then, the media were changed after 2-3 days with fresh media with or without ATRA as previously highlighted in Table 1 and then incubated in a humidified incubator for additional 2-3 days. After 5 days of treatment with ATRA, all experimental groups were washed twice with blank media without any supplementations to remove the FBS and ATRA remnants in the cell culture before the second “BDNF stage” for the ATRA→BDNF and ATRA→0% serum groups. Subsequently, the designed experimental ATRA→BDNF group received BDNF supplementation (78005, STEMCELL TECHNOLOGIES; SRP3014, Sigma-Aldrich, UK) in serum-free media whereas its control group (ATRA→0% serum) received only serum-free media. The ATRA→0% serum group (identical group to ATRA→BDNF but with absence of BDNF) was added to control the presence of BDNF in the ATRA→BDNF group and determine if the absence of BDNF would result in the same outcomes. The other two experimental groups (control and ATRA) were continued culturing in 10% FBS DMEM/F12 media with and without ATRA as previously described. All cell culture groups were incubated in a humidified incubator till the next media change. Finally, the subsequent media change was performed after 3-4 days prior to the end of the neurogenic induction period (12 days). For more details regarding preparation, diluting the differentiating supplements and culturing, see the supplementary S2 file.”

Table 1: Experimental groups (differentiated and control groups)

*Supplemented with penicillin/streptomycin

$DMSO is added as it is the dissolvent used to prepare the ATRA, so the control group is identical to differentiating group but without the differentiating supplement “ATRA”.

Comment 9: Please remove “To the best of our knowledge” and rephrase that statement. At present, the reviewer cannot agree that this is a “relatively simple approach” since the methods are not well described. Also, please remove the word “relatively” from the manuscript. Relatively to what? Also, “relatively” may be denote a personal opinion. Summary: please keep the scientific tone and forego “to our knowledge, relatively, simplified….” Because the context of other protocols is not presented yet.

Response: 

We removed “To the best of our knowledge” and the term “relatively” and modified the statement into the following statement (highlighted in the manuscript, see page 20, 1st paragraph of the discussion section, line 12 to 14).

“This novel and simple approach for establishing a neuronal DPSC differentiation model is supported by microscopic, molecular, and functional evidence."

 

Comment 10: by the ATRA→BDNF protocol: one cannot understand what do the authors mean by the ATRA→BDNF protocol.

Response: 

We clarified this point as follows (highlighted in page 21, line 4-6): 

“Indeed, the differentiated of hDPSCs by the sequential supplementations of ATRA and then BDNF treatment resulted in apparent neuronal morphological features such as phase-bright cell bodies and bipolar or multipolar neurite-like extensions as shown in Fig 1.”

Comment 11: Please state in the methods/results/discussion the specific reasons for using ATRA and BDNF in this work. What are the reasons and expected outcomes? “Atra was used to induce a b and c as reported by xyz…”

Response: 

Thank you for this comment. We added the following paragraph to the discussion section (highlighted with yellow background, see page 20, line 16 to page 21, line 3):

“The ATRA is commonly used to induce neurogenic differentiation of multiple cell lines and stem cells such as SH-SY5Y human neuroblastoma cells [107, 108], P19 mouse embryonal carcinoma cell line [109, 110], embryonic stem cells [111, 112], and mesenchymal stem cells [113]. However, Takahashi et al. [114] and Goldie et al., [74] reported that ATRA alone results in immature neural differentiation of SH-SY5Y and neural stem cells and should be supplemented in combination with neurotrophin such as BDNF to establish full neural maturation. In this context, Encinas et al., [72] and Takahashi et al. [114] reported that sequential induction of ATRA and followed by neurotrophin treatment is critical as the ATRA pretreatment increases the cellular response to neurotrophin (s), including BDNF in SH-SY5Y cells and neural stem cells, respectively. Another study by Bi et al., [115] emphasized that the ATRA pre-induction “activating retinoid signaling” improved neural differentiation of mesenchymal stem cells. Hence, it was hypothesized that the successful neuronal differentiation by sequential supplementation of ATRA and then BDNF in SH-SY5Y cells would also result in mature neuronal differentiation of hDPSCs.”

Comment 12: Apparent neuronal morphological features: please describe the features.

Response: 

Thank you for the comment. The apparent neuronal morphological features observed in the differentiated hDPSCs have now been further described (highlighted with yellow background in the manuscript, page 21, line 5 to 6):

“Indeed, the differentiated of hDPSCs by the sequential supplementations of ATRA and then BDNF treatment resulted in apparent neuronal morphological features such as phase-bright cell bodies and bipolar or multipolar neurite-like extensions as shown in Fig 1.”

Comment 13: Going back the results the reviewer is confronted by “BDNF and FBS supplementation (ATRA→0% serum)” but again, the methods and fig 1 do not show evidently how the work was performed. It becomes very difficult to review the work as it is up to the reviewer/reader to understands the arrangement and rationale for the controls, additional groups… The authors need to describe the work in full. Also, include description (in methods, results, or discussion) for some of the choices. For instance “we added the group ATRA→0% serum to evaluate this and that” or “the group ATRA→0% serum was added to control the presence of something…” This seems to be a well executed work, but it is not possible to understand the procedures and rationale for many of the groups tested/information presented. It may make sense for the authors, but for those reading the work for the first time, it is nearly impossible to understand rationales/procedures.

Response: 

We have now added details regarding the groups in the methods’ section “Neurogenic differentiation of hDPSCs” and also in the supplementary S2 file to describe the work in full. Regarding the addition ATRA→0% serum group was clarified as the following (highlighted in page 8, line 15 to 21) 

“Subsequently, the designed experimental ATRA→BDNF group received BDNF supplementation (78005, STEMCELL TECHNOLOGIES; SRP3014, Sigma-Aldrich, UK) in serum-free media whereas its control group (ATRA→0% serum) received only serum-free media. The ATRA→0% serum group (identical group to ATRA→BDNF but with absence of BDNF) was added to control the presence of BDNF in the ATRA→BDNF group and determine if the absence of BDNF would result in the same outcomes.”

Comment 14: heterogeneity of specific markers in hDPSC cultures which results in the guiding of the cells towards specific neuronal lineages: please explain. What do the authors mean about heterogeneity of specific markers and which are the “specific lineages”. Please note that it is up to the reader to read papers 53-54 to understand this statement.

Response: 

This statement is further explained as follows (highlighted in the manuscript, page 21, line 23 to page 22, line 6):

“In other words, hDPSC cultures have differences in cellular markers that govern the differentiation ability toward a specific cell type. For example, the DPSCs expressing high nestin were reported to differentiate into neuronal and glial lineages compared with no differentiation of DPSCs with low nestin expression [117]. In this regard, we assume that there are additional specific makers/factors which may induce the individual cell populations in the same hDPSC cultures to differentiate into bipolar or multipolar morphology. Thus, further investigation is required to identify the reason(s) underlying the different outcomes of differentiated neuronal-like cells in the same cell culture induced by the same inducers.”

Comment 15: relatively small population: remove the word “relatively” from the manuscript and use objective terms.

Response: 

We have removed this word completely from the manuscript.

Comment 16: low percentage were characterized as stem cells, however, they had not been cell-sorted resulting in low percentage of stem cells in these DPSC cultures: this seems to be a too convenient explanation. If the authors used DPSC (although not all are DPSC), then blaming on cell sorting is very strange. Either the authors used DPSC or not. If cell isolation and characterization were enough to confirm these were dpsc (and the authors used them!)… it is not acceptable to read that, maybe, only a small percentage was dpsc. If this is the case, the basis of this work is compromised and this is not a work on neurogenic differentiion of DPSC. Please explain this contradiction.

Response: 

Thank you for your comment. We agree that our previous explanation regarding cell sorting is not appropriate. We have now modified this text as follows (highlighted in the manuscript, page 22, line 9 to 15):

“Similarly, Kiraly et al. [40] reported functional typical neuronal-like cells differentiated from hDPSCs, however they are a small proportion compared with the entire cell culture and the authors interpreted that it may be because of the high proliferative capacity of the undifferentiated early hDPSC passages. This may be one explanation for the presence of small typical neuronal-like population in our study as we used early passage hDPSCs. Another explanation may be that there is a variation in the response of the hDPSCs to neurogenic induction due to the inherent heterogeneity of DPSCs [117].”

Comment 17: The TUBB3 expression has previously been reported in undifferentiated stem cells, including DPSCs. Please explain the relevancy of such marker. Just mentioning that it has been previously observed does not add much value to the discussion.

Response: 

We have now added an explanation for the relevance of the TUBB3 marker as follows (highlighted in the manuscript, page 22, line 17 to 24):

“TUBB3 is a cytoskeletal protein of neuronal cells and is required in neurodevelopment for guidance, differentiation, survival of neuronal cells [96, 119] and axonal regeneration [120]. This marker has been considered as a specific marker for neuronal cells and widely used in neuronal differentiation studies. However, ours and other studies have detected TUBB3 expression in undifferentiated DPSCs [68, 121]. TUBB3 has also been found in non-neuronal cells such as tumor cells and normal cells such as fibroblast, stroma cells, endocrine cells [122], and perivascular cells, including smooth muscle, and pericytes [123].”

Comment 18: Please explain the most relevant markers observed in your work. Do note that the reader may not understand fully the relevancy of markers such as Tub3, Chat and others. Hence, discuss your findings and the roles of ACHE and CHAT on DPSC neurogenic differentiation/functions in the lights of - 10.1016/j.jfma.2014.09.003 - 10.22203/eCM.v041a16 - 10.1002/jcp.24570 - 10.1002/jcp.25314

Response: 

Thank you for providing the additional references. The relevancy of markers has now been described and explained as follows: 

Cholinergic markers (highlighted in the manuscript, page 23, line 6 to 15):

“For instance, the ATRA→BDNF group demonstrated gene upregulation of cholinergic neurotransmitter markers: choline acetyltransferase (CHAT) and acetylcholinesterase (ACHE) which are responsible for synthesis of neurotransmitter acetylcholine and modulation and termination of synaptic transmission function of neurotransmitter acetylcholine at postsynaptic cholinergic junction, respectively [124-126]. Simultaneously, the ATRA→BDNF group showed no change or absence of detection of the noradrenergic marker (DBH) [104] which is responsible for production of the norepinephrine neurotransmitter for synaptic transmission function of noradrenergic neurons. This upregulation of cholinergic markers combined with no change or absence of noradrenergic marker suggest a specialized cholinergic identity of the resultant neuronal-like cells derived from hDPSCs and SH-SY5Y cells.”

Sensory markers (highlighted in the manuscript, page 23, line 25 to page 24, line 2):

“The POU4F1/BRN3A plays a neurodevelopmental role for sensory neurons [132] whereas SCN9A/Nav 1.7 is responsible for pain sensation detected by sensory neurons [133].”

Note: The provided “10.1016/j.jfma.2014.09.003” paper used the cholinergic marker “ChAT” as marker for motor neuron and this is not scientifically sound as ChAT is not restricted to motor neurons. The Acetyl choline is the most common neurotransmitter in the nervous system and has been found in motor and sensory neurons (see below papers). Also, they are different classifications of neurons according to neurotransmitter type “cholinergic, adrenergic, dopaminergic…” and according to function “motor or sensory” which cannot be connected.

10.3390/ijms22115499

10.1016/j.lfs.2012.08.026

10.1038/s41598-021-96696-8

Comment 19: Please consider having easier names to your groups… what shall the reviewer understand about hDPSC ATRA→BDNF group. Is this the name of the group? Does it mean that hDPSC treated with ATRA and subsequently with BDNF? This goes back to the comments about the methods, it is so unclear that it becomes very difficult to even assess the findings. This is a major concern for this paper. Do not that so far very few comments are given to the results because it is nearly impossible to understand how this work was performed (hence the data and controls cannot be even assessed at this point).

Response: 

We apologise for any confusion caused. The ATRA→BDNF group is the name of the group wherein “hDPSCs were treated with ATRA and subsequently with BDNF”. Each group name and their descriptions: the media, supplements, one or two stages, and duration time (days) are now clarified in Table 1, page 9. In addition, detailed description is found in the supplementary S2 file. 

Comment 20: The possible explanation for this is that the sonic hedgehog: this statement seems to be incomplete.

Response: 

Apologies for this, the statement is now modified as following (highlighted in the manuscript, page 24, line 8 to 11).

“The possible explanation for motor neuronal differentiation in these studies is the presence of the sonic hedgehog (SHH) supplementation in the differentiating media which is reported as an active inducer for guided differentiation into the motor neuronal lineage [134, 135].”

Comment 21: addition, their use: should be the use

Response: 

We have modified this accordingly (highlighted in the manuscript, page 24, line 11 to 14):

“In addition, the use of very low concentrations of ATRA (1nM to 2μM) and BDNF (10-20 ng/ml) in these studies may not be sufficient to guide the differentiation toward a sensory neuronal lineage in comparison with the current study (10μM ATRA and 50 ng/ml BDNF).”

Comment 22: into sensory cholinergic neuronal-like cells: this seems to be an overestimation of the results. Have the authors measured choline release? The reviewer understand that the markers are present, but this is a hallmark of choline production and release. Please rephrase or highlight the evidence that confirms that these are functional cholinergic neurons.

Response: 

We thank the reviewer for raising this point. We believe that a combination of electrophysiological recordings (which reflect the functionality), and the expression of specific sets of neurogenic lineage gene markers would be a strong basis for our conclusion. We have now rephrased the text as follows: (highlighted with yellow background in the manuscript, page 26, line 13 to 14):

“In conclusion, this study provides original evidence for differentiation of human DPSCs into neuronal-like cells, particularly toward sensory cholinergic neuronal cells.”

Overall, this paper has a lot of potential, but the writing needs to be much improved. The reader is left alone to understand the procedures and relevancy of the work. At this stage, it is not possible to discuss the data in depth because it is not clear how the work was done, which controls were employed (it is true that the authors mention, but the reader shall not make all the effort to understand how this work was done). Hence, the reviewer cannot analyze the data without knowing for sure how that work was done and which group served as a control for what… The authors are invited to submit a revised version that will allow the reviewer to be sure that everything is on place. It is a nice paper but needs to be more self-contained.

Response: 

Thank you for the opportunity to modify and strengthen our manuscript. As described above we have provided multiple amendments which we feel now significantly enhance the presentation of the study.

---

## [Decision Letter · Decision Letter 1]

21 Oct 2022

Exploring the neurogenic differentiation of human dental pulp stem cells

PONE-D-22-06526R1

Dear Dr. Al-Maswary,

We’re pleased to inform you that your manuscript has been judged scientifically suitable for publication and will be formally accepted for publication once it meets all outstanding technical requirements.

Kind regards,

Michal Hetman

Academic Editor

PLOS ONE

Additional Editor Comments (optional):

Reviewers' comments:

Reviewer's Responses to Questions

**Comments to the Author**

1. If the authors have adequately addressed your comments raised in a previous round of review and you feel that this manuscript is now acceptable for publication, you may indicate that here to bypass the “Comments to the Author” section, enter your conflict of interest statement in the “Confidential to Editor” section, and submit your "Accept" recommendation.

Reviewer #1: All comments have been addressed

Reviewer #2: All comments have been addressed

2. Is the manuscript technically sound, and do the data support the conclusions?

Reviewer #1: Yes

Reviewer #2: Yes

3. Has the statistical analysis been performed appropriately and rigorously? 

Reviewer #1: Yes

Reviewer #2: Yes

4. Have the authors made all data underlying the findings in their manuscript fully available?

Reviewer #1: Yes

Reviewer #2: Yes

5. Is the manuscript presented in an intelligible fashion and written in standard English?

Reviewer #1: Yes

Reviewer #2: Yes

6. Review Comments to the Author

Reviewer #1: All my concerns have been addressed. Reviewer have no more question. It could be accepted for publication.

Reviewer #2: (No Response)

7. PLOS authors have the option to publish the peer review history of their article (what does this mean?). If published, this will include your full peer review and any attached files.

Reviewer #1: **Yes: **Zhi Chen

Reviewer #2: No

---

## [Editor Report · Acceptance letter]

28 Oct 2022

PONE-D-22-06526R1 

Exploring the neurogenic differentiation of human dental pulp stem cells 

Dear Dr. Al-Maswary:

I'm pleased to inform you that your manuscript has been deemed suitable for publication in PLOS ONE. Congratulations! Your manuscript is now with our production department. 

Kind regards, 

on behalf of

Dr. Michal Hetman 

Academic Editor

PLOS ONE